

# Interspecific variation in the limb long bones among modern rhinoceroses—extent and drivers

Christophe Mallet[1], Raphaël Cornette[2], Guillaume Billet[3] and Alexandra Houssaye[1]

[1] Mécanismes adaptatifs et évolution (MECADEV), UMR 7179, MNHN, CNRS, Muséum National d'Histoire Naturelle, Paris, France
[2] Institut de Systématique, Evolution, Biodiversité (ISYEB), UMR 7205, MNHN, CNRS, SU, EPHE, UA, Muséum National d'Histoire Naturelle, Paris, France
[3] Centre de Recherche en Paléontologie—Paris (CR2P), UMR CNRS 7207, MNHN, CNRS, SU, Muséum National d'Histoire Naturelle, Paris, France

Corresponding author
Christophe Mallet,
christophe.mallet@edu.mnhn.fr

## ABSTRACT

Among amniotes, numerous lineages are subject to an evolutionary trend toward body mass and size increases. Large terrestrial species may face important constraints linked to weight bearing, and the limb segments are particularly affected by such constraints due to their role in body support and locomotion. Such groups showing important limb modifications related to high body mass have been called "graviportal." Often considered graviportal, rhinoceroses are among the heaviest terrestrial mammals and are thus of particular interest to understand the limb modifications related to body mass and size increase. Here, we present a morphofunctional study of the shape variation of the limb long bones among the five living rhinos to understand how the shape may vary between these species in relation with body size, body mass and phylogeny. We used three dimensional geometric morphometrics and comparative analyses to quantify the shape variation. Our results indicate that the five species display important morphological differences depending on the considered bones. The humerus and the femur exhibit noticeable interspecific differences between African and Asiatic rhinos, associated with a significant effect of body mass. The radius and ulna are more strongly correlated with body mass. While the tibia exhibits shape variation both linked with phylogeny and body mass, the fibula displays the greatest intraspecific variation. We highlight three distinct morphotypes of bone shape, which appear in accordance with the phylogeny. The influence of body mass also appears unequally expressed on the different bones. Body mass increase among the five extant species is marked by an increase of the general robustness, more pronounced attachments for muscles and a development of medial parts of the bones. Our study underlines that the morphological features linked to body mass increase are not similar between rhinos and other heavy mammals such as elephants and hippos, suggesting that the weight bearing constraint can lead to different morphological responses.

## INTRODUCTION

Many vertebrate lineages exhibit convergence toward a body mass increase through time (*Depéret, 1907*; *Raia et al., 2012*; *Baker et al., 2015*; *Bokma et al., 2016*). Size and mass augmentation implies metabolic and musculoskeletal modifications for the whole body to bear its own weight (*McMahon, 1973*). One of the most noticeable body changes related to weight bearing concern modifications of the appendicular skeleton; animals displaying such adaptive traits are said to be "graviportal" (*Hildebrand, 1974*). This concept introduced by *Gregory (1912)* and *Osborn (1929)* has been defined based on both anatomical and locomotion aspects: the commonly accepted criteria are, in addition to a body mass of several hundreds of kilograms, columnar limbs with stylopodium lengthening and autopodium shortening, robust bones (i.e., larger shaft for a given length), large feet with enlarged adipose cushions, reduced phalanges, long strides associated with the inability to gallop (*Gregory, 1912*; *Osborn, 1929*; *Coombs, 1978*). This condition was opposed to the "cursorial" one characterizing running animals (e.g., horses and many ungulates). Between these two extremes, intermediate categories tended to sharpen this tentative locomotor classification, with "subcursorial" for moderate cursorial adaptations with good running performances (e.g., felids and canids), and "mediportal" for animals with conformations meeting both the weight bearing aspect and running capacities (e.g., suids, tapirs) (*Gregory, 1912*; *Coombs, 1978*; *Eisenmann & Guérin, 1984*). These categories remain extensively used in functional morphology and locomotion studies (*Maynard Smith & Savage, 1956*; *Coombs, 1978*; *Eisenmann & Guérin, 1984*; *Prothero, Manning & Hanson, 1986*; *Biewener, 1989a*; *Stein & Casinos, 1997*; *Polly, 2007*; *Scherler et al., 2013*; *MacLaren & Nauwelaerts, 2016*). *Hildebrand (1974)* proposed an arbitrary body mass of 900 kg beyond which the species is considered as graviportal, but without justification for this threshold. *Carrano (1999)* tackled this problem by replacing these discrete categories by a multivariate continuum of locomotor habits ranging from graviportal to cursorial based on bone and muscular insertion measurements, chosen to be "biomechanically relevant" but performed only on the femur, tibia and third metatarsal.

As a consequence, the categorization of some taxa as graviportal may vary depending on authors. Among living mammals, elephants, rhinos and hippos are commonly considered as the three main graviportal taxa (*Alexander & Pond, 1992*). Elephants obviously fulfill all the morphological and biomechanical criteria defining graviportality (*Coombs, 1978*; *Langman et al., 1995*). However, the peculiar morphology of hippos (barrel-like body and shortened limbs) linked to semi-aquatic habits (*Mazza, 2014*) has been considered alternately as mediportal (*Coombs, 1978*; *Ross, 1984*) or graviportal (*Alexander & Pond, 1992*; *Carrano, 1999*; *MacFadden, 2005*; *Stilson, Hopkins & Davis, 2016*). The graviportal condition in rhinoceroses is surely the least consensual: *Gregory (1912)* and *Osborn (1929)* considered rhinos as mediportal whereas later works assigned them a graviportal condition (*Prothero & Sereno, 1982*; *Eisenmann & Guérin, 1984*). *Becker (2003)* and *Becker et al. (2009)* dug into this question and developed a "gracility index" based on the work of *Guérin (1980)* to categorize modern and fossil rhinos, but only based on third metacarpal and metatarsal proportions. The use of this index refined the
**Table 1  Main characteristics of the five studied species.**

| Species name | Total body length (cm) | Shoulder height (cm) | Mean body mass (kg) | Ecology | Locomotor type | | |
|---|---|---|---|---|---|---|---|
| | | | | | *Gregory (1912), Osborn (1929),* and *Coombs (1978)* | *Eisenmann & Guérin (1984)* | *Becker (2003)* |
| *Ceratotherium simum**\** | 340–420 | 150–180 | 2,300 | Open savanna | M | G | G |
| *Dicerorhinus sumatrensis**\*\** | 236–318 | 100–150 | 775 | Dense forests and swampy lakes | M | G | M |
| *Diceros bicornis**\** | 300–380 | 140–170 | 1,050 | Open savanna and clear forest | M | G | M |
| *Rhinoceros sondaicus**\*\** | 305–344 | 150–170 | 1,350 | Dense forests and swampy areas | M | G | G |
| *Rhinoceros unicornis**\*\** | 335–346 | 175–200 | 2,000 | Floodplains and swamps | M | G | M |

Notes:
Length, height and body mass data compiled and calculated after *Dinerstein (2011)*. Shoulder height is given at the withers. Ecological data compiled after *Becker (2003)*.
G, graviportal; M, mediportal.
\* African species.
\*\* Asiatic species.

classification of modern rhinos distinguishing mediportal and graviportal species instead of a single class attribution for the whole family (Table 1).

Regardless of the locomotor type to which they belong, the family Rhinocerotidae includes some of the heaviest land mammal species after elephants, displaying adaptations to support their high body mass (*Alexander & Pond, 1992*). There are five remaining species of rhinos on Earth nowadays: the White Rhinoceros (*Ceratotherium simum* Burchell, 1817) and the Black Rhinoceros (*Diceros bicornis* Linnaeus, 1758) both live in sub-Saharan Africa, whereas the Indian Rhinoceros (*Rhinoceros unicornis* Linnaeus, 1758), the Javan Rhinoceros (*R. sondaicus* Desmarest, 1822) and the Sumatran Rhinoceros (*Dicerorhinus sumatrensis* Fischer, 1814) survive in India and Nepal, Java and Sumatra, respectively (*Dinerstein, 2011*). These species exhibit an important variation in body mass and size (Table 1), ranging from less than a ton for *Dicerorhinus sumatrensis* to more than three tons for the biggest known specimens of *C. simum*. They are all good walkers and runners, able to gallop and reach an elevated speed (27 km/h for *C. simum*, *Alexander & Pond, 1992*; 45 km/h for *Diceros bicornis*, *Blanco, Gambini & Fariña, 2003*). However, important ecological differences also exist (*Groves, 1967a, 1967b, 1972*; *Groves & Kurt, 1972*; *Laurie, Lang & Groves, 1983*; *Hillman-Smith & Groves, 1994*; *Dinerstein, 2011*; *Groves & Leslie, 2011*): the three Asiatic rhinos are excellent swimmers and very familiar with an aquatic environment whereas the two African ones are easily stopped by a relatively deep river (*Guérin, 1980*). While *C. simum* is a pure grazer, *R. unicornis* can both graze and browse small shrubs, leafy material and fruits, the three other species being mainly leaf browsers. Before the drastic decrease of their natural habitats under human pressure, rhinos occupied a wide geographic range across Africa and Asia (*Dinerstein, 2011*; *Rookmaaker & Antoine, 2013*). Moreover, the fossil record of the superfamily Rhinocerotoidea contains many lineages displaying evolutionary convergence toward an increase of body mass (*Prothero & Schoch, 1989*; *Prothero, 1998*;

*Antoine, 2002*; *Becker, 2003*; *Scherler et al., 2013*). However, despite the importance of rhino species for understanding evolution toward large body mass and the fact that they are some of the heaviest surviving land mammals, only a few studies really explore the variation of their limb bone morphology in relation to their body proportions (*Guérin, 1980*; *Eisenmann & Guérin, 1984*). After the pioneering works of *Cuvier (1812)* and *De Blainville & Nicard (1839)* describing the postcranial anatomy of modern rhinos, almost no work tried to broadly analyze and compare the morphology of their limb bones. *Guérin (1980)* published a substantial comparative anatomy work on the whole skeleton of the five extant species. This study emphasized anatomical descriptions with a direct application on the determination of fossil forms. Despite considerations on inter- and intraspecific osteological variation in modern rhinos, this work did not fully explore the patterns of shape variation in this group. Furthermore, most of the previous studies used a classic morphometric approach with linear measurements on bones, an approach which cannot precisely take into consideration the whole shape of the bone in three dimensions (3D). To our knowledge, no morphofunctional analyses have been carried out on limb long bones of modern rhinos taking into consideration their whole shape.

Here, we hypothesize that modern rhinoceroses exhibit a large amount of interspecific variation of the shape of each bone that would be essentially associated with a strong effect of body mass on bone morphology. We predict that this effect will be more pronounced on the stylopodium (humerus and femur) than on the zeugopodium (radius, ulna, tibia and fibula) bones. This would be in accordance with previous works on changes of limb shape between graviportal and cursorial taxa (*Biewener, 1989b*; *Campione & Evans, 2012*). In addition, we expect an effect of phylogenetic heritage and different species' ecologies on bone shape. To test these hypotheses, we propose to explore the variation in the shape of the limb long bones among the five modern rhino species using a 3D geometric morphometrics approach. We describe interspecific patterns of morphological variation for the six bones composing the stylopodium and the zeugopodium, taking into account the intraspecific variation.

## MATERIALS AND METHODS

### Sample

We selected 62 dry skeletons in different European museums belonging to the five extant rhino species: *C. simum*, *Dicerorhinus sumatrensis*, *Diceros bicornis*, *R. sondaicus* and *R. unicornis* (Table 2). We followed the taxonomic attribution given by each institution for most of the specimens, except for three individuals determined or reattributed by ourselves on osteological criteria and later confirmed by our morphometric analysis (see Table 2). We studied altogether 53 humeri, 49 radii, 46 ulnae, 56 femora, 52 tibiae and 50 fibulae, with 37 skeletons being complete. We included only mature specimens with fully fused epiphyses (adults) or specimens where the line of the epiphyseal plates was still visible on some bones (subadults). Bones showing breakages or unnatural deformations were not considered in our analysis. It has been proved that feet bones are subject to important osteopathologic deformations in rhinos (*Regnault et al., 2013*). However, in accordance with the observations of *Guérin (1980)*, we did not notice any major difference between the

**Table 2 List of the studied specimens with skeletal composition, sex, age class, condition and 3D acquisition details.**

| Taxon | Institution | Specimen number | H | R | U | Fe | T | Fi | Sex | Age | Condition | 3D acquisition |
|---|---|---|---|---|---|---|---|---|---|---|---|---|
| *Ceratotherium simum** | NHMUK | ZD 2018.143 | X | X | X | X | X | X | U | A | U | SS |
| *Ceratotherium simum* | NHMW | 3086 | X | X | X | X | X | X | U | A | W | P |
| *Ceratotherium simum* | RBINS | 19904 | X | X | X | X | X | X | M | S | W | SS |
| *Ceratotherium simum* | RBINS | 35208 | X | X | X | X | | X | U | A | U | SS |
| *Ceratotherium simum* | RMCA | 1985.32-M-0001 | X | X | X | X | X | X | U | A | W | SS |
| *Ceratotherium simum* | RMCA | RG35146 | X | X | X | X | X | X | M | A | W | SS |
| *Ceratotherium simum* | UCMP | 125000 | | | | X | | | U | A | U | CT |
| *Ceratotherium simum* | ZSM | 1912/4199 | | | | X | | | U | A | W | SS |
| *Ceratotherium simum* | BICPC | NH.CON.20 | X | X | X | X | X | X | M | S | W | SS |
| *Ceratotherium simum* | BICPC | NH.CON.32 | X | X | X | X | X | X | F | S | W | SS |
| *Ceratotherium simum* | BICPC | NH.CON.37 | X | X | | X | X | X | F | A | W | SS |
| *Ceratotherium simum* | BICPC | NH.CON.40 | X | X | X | X | X | X | F | S | W | SS |
| *Ceratotherium simum* | BICPC | NH.CON.110 | X | X | X | X | X | X | M | A | W | SS |
| *Ceratotherium simum* | BICPC | NH.CON.112 | X | X | X | X | X | X | M | A | W | SS |
| *Ceratotherium simum* | NMS | NMS.Z.2010.44 | X | | | X | | | F | A | U | CT |
| *Ceratotherium simum* | MNHN | ZM-MO-2005-297 | X | | | X | X | X | M | A | C | SS |
| *Dicerorhinus sumatrensis* | MNHN | ZM-AC-1903-300 | X | X | X | X | X | X | M | A | W | SS |
| *Dicerorhinus sumatrensis* | MNHN | ZM-AC-A7967 | X | X | X | | | | F | A | W | SS |
| *Dicerorhinus sumatrensis* | NHMUK | ZD 1879.6.14.2 | X | X | X | X | X | X | M | A | W | SS |
| *Dicerorhinus sumatrensis* | NHMUK | ZD 1894.9.24.1 | X | X | X | X | X | X | U | A | W | SS |
| *Dicerorhinus sumatrensis* | NHMUK | ZD 1931.5.28.1 | X | X | X | X | X | X | M | S | W | SS |
| *Dicerorhinus sumatrensis* | NHMUK | ZE 1948.12.20.1 | X | X | X | X | X | X | U | A | U | SS |
| *Dicerorhinus sumatrensis* | NHMUK | ZE 1949.1.11.1 | X | X | X | X | X | X | U | A | W | SS |
| *Dicerorhinus sumatrensis* | NHMUK | ZD 2004.23 | X | | | X | X | X | U | A | W | SS |
| *Dicerorhinus sumatrensis* | NHMW | 1500 | | | | X | X | X | M | A | U | P |
| *Dicerorhinus sumatrensis* | NHMW | 3082 | X | X | X | X | X | X | U | A | U | P |
| *Dicerorhinus sumatrensis* | NHMW | 29568 | | X | X | X | | X | U | S | U | P |
| *Dicerorhinus sumatrensis* | RBINS | 1204 | X | X | X | X | X | X | M | A | W | SS |
| *Dicerorhinus sumatrensis* | UMZC | H.6392 | X | | | | | | U | A | U | CT |
| *Dicerorhinus sumatrensis* | ZSM | 1908/571 | X | X | | X | X | Fi | M | A | U | SS |
| *Diceros bicornis* | CCEC | 50002040 | X | | | X | X | X | U | A | W | SS |
| *Diceros bicornis* | CCEC | 50002044 | | X | | X | | | U | S | U | SS |
| *Diceros bicornis* | CCEC | 50002045 | | | | X | | | U | S | W | SS |
| *Diceros bicornis* | CCEC | 50002046 | X | X | X | | X | X | U | S | U | SS |
| *Diceros bicornis* | CCEC | 50002047 | | X | X | | X | X | U | A | U | SS |
| *Diceros bicornis* | MNHN | ZM-AC-1936-644 | X | X | X | X | X | X | F | S | U | SS |
| *Diceros bicornis* | MNHN | ZM-AC-1944-278 | X | | | X | X | X | M | A | C | SS |
| *Diceros bicornis* | MNHN | ZM-AC-1974-124 | | | | X | X | X | F | A | C | SS |
| *Diceros bicornis* | RBINS | 9714 | X | X | X | X | X | X | F | A | W | SS |
| *Diceros bicornis* | RMCA | RG2133 | X | X | X | X | X | X | M | S | W | SS |
| *Diceros bicornis* | UCMP | 9856 | | | | X | | | U | A | U | CT |
| *Diceros bicornis* | ZSM | 1961/186 | X | X | X | X | X | X | M | S | U | SS |

(Continued)

| Taxon | Institution | Specimen number | H | R | U | Fe | T | Fi | Sex | Age | Condition | 3D acquisition |
|---|---|---|---|---|---|---|---|---|---|---|---|---|
| *Diceros bicornis* | ZSM | 1961/187 | X | X | X | X | X | X | M | S | U | SS |
| *Diceros bicornis* | ZSM | 1962/166 | X | X | X | X | X |  | F | S | U | SS |
| *Rhinoceros sondaicus* | CCEC | 50002041 | X | X | X | X | X | X | U | A | W | SS |
| *Rhinoceros sondaicus* | CCEC | 50002043 | X | X | X | X |  |  | U | A | W | SS |
| *Rhinoceros sondaicus* | MNHN | ZM-AC-A7970 | X | X | X | X | X | X | U | A | U | SS |
| *Rhinoceros sondaicus* | MNHN | ZM-AC-A7971 | X | X | X | X | X | X | U | A | W | SS |
| *Rhinoceros sondaicus* | NHMUK | ZD 1861.3.11.1 | X | X | X | X | X | X | U | S | W | SS |
| *Rhinoceros sondaicus* | NHMUK | ZD 1871.12.29.7 | X | X | X | X | X | X | M | A | W | SS |
| *Rhinoceros sondaicus* | NHMUK | ZD 1921.5.15.1 | X | X | X | X | X | X | F | S | W | SS |
| *Rhinoceros sondaicus* | RBINS | 1205F | X | X | X | X | X | X | U | S | W | SS |
| *Rhinoceros unicornis*** | MNHN | ZM-AC-1885-734 | X | X | X | X | X |  | U | A | W | SS |
| *Rhinoceros unicornis* | MNHN | ZM-AC-1932-49 | X |  |  |  | X | X | U | S | U | SS |
| *Rhinoceros unicornis* | MNHN | ZM-AC-1960-59 | X | X | X | X | X | X | M | A | C | SS |
| *Rhinoceros unicornis* | MNHN | ZM-AC-1967-101 | X | X | X | X | X |  | F | A | C | SS |
| *Rhinoceros unicornis* | NHMUK | ZD 1884.1.22.1.2 | X | X | X | X | X | X | F | A | W | SS |
| *Rhinoceros unicornis* | NHMUK | ZE 1950.10.18.5 | X | X | X | X | X | X | M | A | W | SS |
| *Rhinoceros unicornis* | NHMUK | ZE 1961.5.10.1 | X | X | X | X | X | X | M | A | W | SS |
| *Rhinoceros unicornis** | NHMUK | ZD 1972.822 | X | X | X | X | X | X | U | A | U | SS |
| *Rhinoceros unicornis* | RBINS | 1208 | X | X | X | X | X | X | F | A | C | SS |
| *Rhinoceros unicornis* | RBINS | 33382 | X | X | X | X | X | X | U | A | U | SS |

Notes:
Bones—H, humerus; R, radius; U, ulna; Fe, femur; T, tibia; Fi, fibula. Sex: F, female; M, male; U, unknown. Age—A, adult; Sa, sub-adult. Condition—W, wild; C, captive; U, unknown. 3D acquisition—SS, surface scanner; P, photogrammetry; CT, CT-scan. Institutional codes: BICPC, Powell Cotton Museum, Birchington-on-Sea; CCEC, Centre de Conservation et d'Étude des Collections, Musée des Confluences, Lyon; MHNT, Muséum d'Histoire Naturelle de Toulouse, Toulouse; MNHN, Muséum National d'Histoire Naturelle, Paris; NHMUK, Natural History Museum, London; NHMW, Naturhistorisches Museum Wien, Vienna; NMS, National Museums Scotland, Edinburgh; RBINS, Royal Belgian Institute of Natural Sciences, Brussels; RMCA, Royal Museum for Central Africa, Tervuren; UCMP, University of California Museum of Paleontology, Berkeley; UMZC, University Museum of Zoology Cambridge, Cambridge; ZSM, Zoologische Staatssammlung München, Munich.
* Specimens NHMUK ZD 2018.143 and NHMUK ZD 1972.822 were determined by ourselves during the visit of the collections on the basis of morphological observations and measurements on the post-cranial elements. These determinations were later confirmed by our shape analysis.
** The specimen MNHN-ZM-AC-1885-734 was previously determined as *Rhinoceros sondaicus* based on a supposed Javan origin. The observations made on both long bones and tarsal elements led us to consider this individual as an Indian rhino (*Rhinoceros unicornis*). This attribution was later confirmed by our shape analysis.

long bones of captive and wild animals, neither through visual and osteological observations nor in our morphometric analyses; we therefore did not take into account this parameter. Sexual dimorphism occurs among rhinos but has been mostly investigated regarding the external morphology of the animals (*Dinerstein, 1991*, *2011*; *Berger, 1994*; *Zschokke & Baur, 2002*). The few studies that have explored the osteological variations between sexes indicated only slight absolute metric divergences depending on species (*Guérin, 1980*; *Groves, 1982*). This suggests that intraspecific variation due to sex may be marginal when compared to interspecific variation, and probably more related to the size of the bone than to the shape. Furthermore, since almost half of our sample lacked sex information and that we had twice as many males than females, we could not carefully address sex in our study (see Results).

## 3D models
Bones were mostly digitized with a structured-light three-dimensional scanner (Artec Eva) and reconstructed with Artec Studio Professional software (v12.1.1.12—*Artec 3D, 2018*).

Complementarily, 19 bones were digitized with a photogrammetric approach, following *Mallison & Wings (2014)* and *Fau, Cornette & Houssaye (2016)*. Sets of photos were taken all around the bones and aligned to reconstruct a 3D model with Agisoft Photoscan software (v1.4.2—*Agisoft, 2018*). Previous studies indicated no significant difference between 3D models obtained with these two methods (*Petti et al., 2008*; *Remondino et al., 2010*; *Fau, Cornette & Houssaye, 2016*). Five bones were digitized using medical computed tomography scanners at the Royal Veterinary College, London (Equine Hospital) and at the University of California, San Francisco (Department of Radiology & Biomedical Imaging). Bone surfaces were extracted as meshes using Avizo software (v9.5.0—*Thermo Fisher Scientific, 2018*). Each mesh was decimated to reach 250,000 vertices and 500,000 faces using MeshLab software (v2016.12—*Cignoni et al., 2008*). We mainly selected left bones during acquisition; when this was impossible, right bones were selected and then mirrored before analysis.

## Anatomical terminology

All anatomical terms used to describe bones come from classic references: the *Nomina Anatomica Veterinaria* (*World Association of Veterinary Anatomists, International Committee on Veterinary Gross Anatomical Nomenclature, 2005*) and anglicized terms of *Barone (2010a)* for general osteology and bone orientation, *Guérin (1980)* for specific rhino anatomy, complemented by the contributions of *Colyn (1980)*, *Antoine (2002)* and *Heissig (2012)*. Despite these previous works, one anatomical feature remained unnamed, leading us to use our own designation: we called "palmar process" the process facing the coronoid process on the palmar border of the radius proximal epiphysis. Muscle insertions were described after the general anatomy of horses (*Barone, 2010b*), complemented by the work of *Beddard & Treves (1889)* and some complementary information from *Guérin (1980)* on rhino myology, *Bressou (1961)* on that of tapirs and *Fisher, Scott & Naples (2007)* and *Fisher, Scott & Adrian (2010)* on that of hippos.

## Geometric morphometrics

To analyze shape variation in our sample, we performed 3D geometric morphometrics, a widely used approach allowing quantification of morphological differences between objects using landmark coordinates (*Adams, Rohlf & Slice, 2004*; *Zelditch et al., 2012*).

### Landmark digitization

Following the procedure described by *Gunz, Mitteroecker & Bookstein (2005)*, *Gunz & Mitteroecker (2013)* and *Botton-Divet et al. (2016)*, we defined the bones' shape using anatomical landmarks and curve and surface sliding semi-landmarks. Each curve is bordered by anatomical landmarks as recommended by *Gunz & Mitteroecker (2013)*. We placed all landmarks and curves using the IDAV Landmark software (v3.0—*Wiley et al., 2005*). We used 35 anatomical landmarks on the humerus, 23 on the radius, 21 on the ulna, 27 on the femur, 24 on the tibia and 12 on the fibula. Details of landmark numbers and locations used for each bone are given in Table S1 and Fig. S1.

Following the procedure detailed by *Botton-Divet et al. (2016)*, we created a template to place surface semi-landmarks for each bone: a specimen (*C. simum* RMCA

1985.32-M-0001) was randomly chosen on which all anatomical landmarks, curve and surface sliding semi-landmarks were placed. We then used this template for the projection of surface sliding semi-landmarks on the surface of the other specimens. Projection was followed by a relaxation step to ensure that projected points matched the actual surface of the meshes. Curve and surface sliding semi-landmarks were then slid to minimize the bending energy of a thin plate spline (TPS) between each specimen and the template at first, and then two times between the result of the preceding step and the Procrustes consensus of the complete dataset. Therefore, all landmarks can be treated at the end as geometrically homologous (*Gunz, Mitteroecker & Bookstein, 2005*) and analyzed with classic procedure such as generalized Procrustes analysis (GPA; see below). Projection, relaxation and sliding processes were conducted using the Morpho package in the R environment (*R Development Core Team, 2014*). Details of the process are provided in the documentation of the package (*Schlager, 2018*).

### Repeatability tests

For each bone, we tested the repeatability of the anatomical landmark digitization taking measurements ten times on three specimens of the same species, *C. simum*, chosen to display the closest morphology and size. We superimposed these measurements using a GPA and visualized the results using a principal component analysis (PCA). Results showed a variation within specimens clearly smaller than the variation between specimens (see Fig. S2) and allowed us to consider our anatomical landmarks as precise enough to describe shape variation.

### Generalized Procrustes Analyses

After the sliding of all semi-landmarks, we performed GPA (*Gower, 1975*; *Rohlf & Slice, 1990*) to remove the effects of size and of the relative position of the points and to isolate only the shape information. As our dataset contained more variables than observations, we used a PCA to reduce dimensionality as recommended by *Gunz & Mitteroecker (2013)* and visualize the specimen distribution in the morphospace. We computed theoretical consensus shape of our sample and used it to calculate a TPS deformation of the template mesh. We then used this newly created consensus mesh to compute theoretical shapes associated with the maximum and minimum of both sides of each PCA, as well as mean shapes of each bone for each species. GPA, PCA and shape computations were done using the "Morpho" and "geomorph" packages (*Adams & Otárola-Castillo, 2013*; *Adams, Collyer & Kaliontzopoulou, 2018*; *Schlager, 2018*) in the R environment (*R Development Core Team, 2014*). Neighbor Joining method was used to construct trees based on relative Euclidian distances between individuals based on all principal component scores obtained with the PCA, allowing a global visualization of the relationships between all the specimens. Trees were computed with the "ape" package (*Paradis, Claude & Strimmer, 2004*).

### Allometry effect

We tested the effect of allometry, defined as "the size-related changes of morphological traits" (*Klingenberg, 2016*). Pearson's correlation tests were performed to look for correlation between the principal components and the centroid size ($\log_{10}$) for each bone.
We also used the function *procD.allometry* of the "geomorph" package to perform a Procrustes ANOVA (a linear regression model using Procrustes distances between species instead of covariance matrices—see *Goodall, 1991*) to quantify the shape variation related to the centroid size, and to visualize theoretical shapes associated with minimal and maximal sizes of our sample (*Adams & Otárola-Castillo, 2013*; *Adams, Collyer & Kaliontzopoulou, 2018*). This test was performed taking into account group affiliation (e.g., species) to highlight respective roles of centroid size and species determination on the shape variation. In the absence of individual body mass for the majority of our sample, we also performed a Procrustes ANOVA with the cube root of the mean mass attributed to each species (Table 1), each specimen being associated with the mean mass of its species. As for the centroid size, theoretical shapes associated with minimal and maximal mean mass were computed using the predicted Procrustes residuals (details on the procedure are given in the "geomorph" documentation). Plots of the multivariate regressions of shape scores (i.e., regression of shape on size; see *Drake & Klingenberg, 2008*) against log-transformed centroid size were also computed.

# RESULTS

## Shape analysis

We describe here the results of our PCA for each bone and focus on the theoretical shape variations along the two main axes. For each bone, we chose to represent relevant views and anatomical features. Complete visualizations of the different theoretical shapes for the two first axes are available in Fig. S3. Analysis of shape relations among our sample is completed by the Neighbor Joining trees provided in Fig. S4.

### Humerus

The first two axes of the PCA computed on the humerus represent 60.6% of the total variance (Fig. 1A). The first axis represents more than half of the global variance (53%) and the five species appear clearly sorted along it, opposing *Dicerorhinus sumatrensis* on the positive side to *C. simum* on the negative one, i.e., the lightest and heaviest species, respectively. *Diceros bicornis* is grouped with *C. simum* on the negative part of the axis, whereas *R. sondaicus* is on the positive part. *R. unicornis* occupies the center of the axis, between *Diceros bicornis* and *R. sondaicus*. Points distribution in the morphospace and Neighbor Joining trees indicate a clear separation between African and Asiatic rhinos (Fig. S4A). The theoretical shape at the PC1 minimum (Figs. 1B, 1D, 1F and 1H) shows a massive morphology, with mediolaterally and craniocaudally broad epiphyses and shaft; a wide humeral head, with very little overhanging of the diaphysis in the caudal direction; a lesser tubercle more strongly developed than the greater tubercle, with an intermediate tubercle separating a widely open bicipital groove into unequal parts, the lateral one being the largest; a lesser tubercle convexity medially extended whereas the greater tubercle one is quite reduced in this direction; a broad and diamond-shaped *m. infraspinatus* imprint on the lateral side; a broad deltoid tuberosity not extending beyond the lateral border of the bone; a shaft with its maximal width situated between the humeral head and the deltoid tuberosity; a distinct but very smooth and flat *m. teres major*

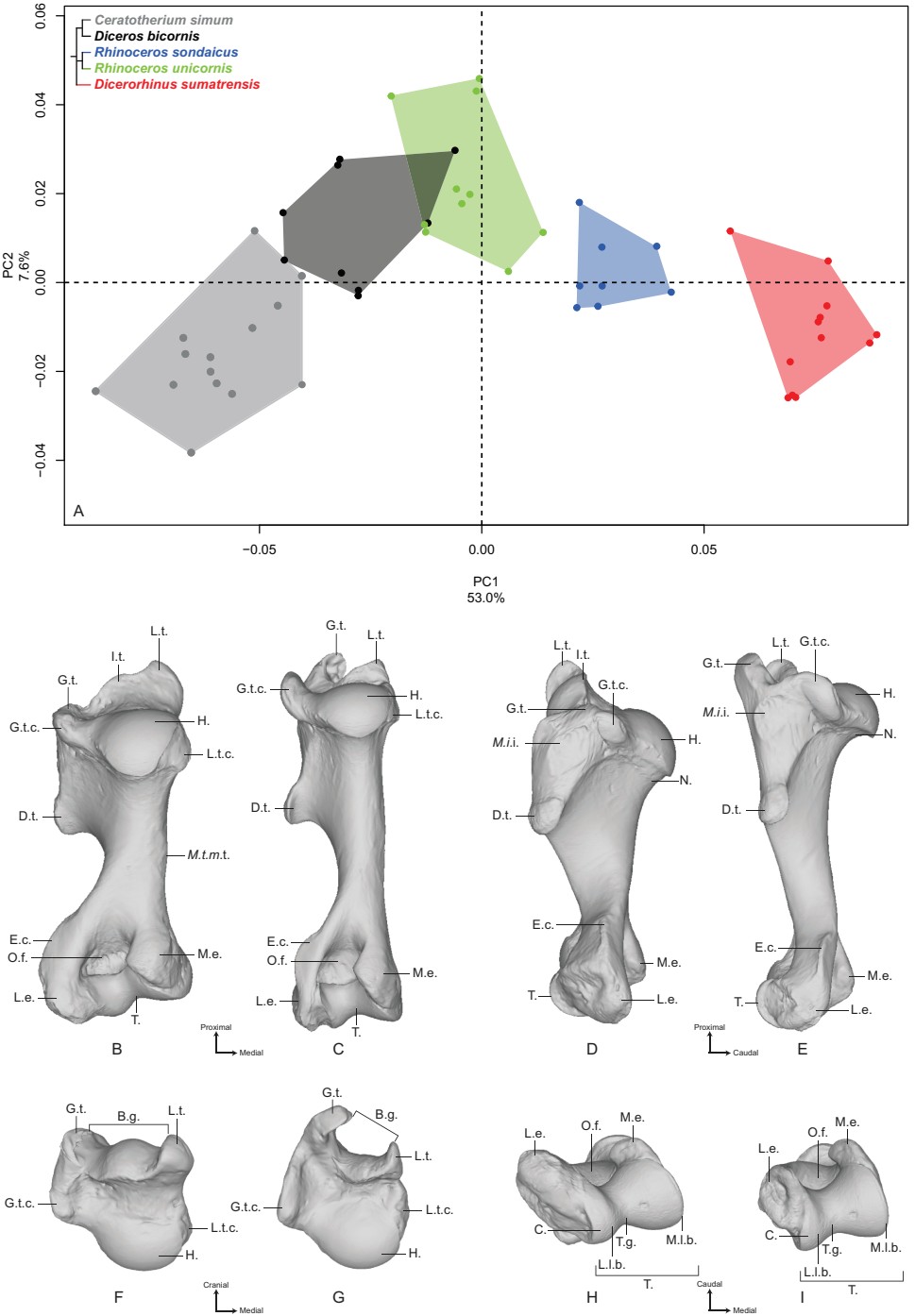

**Figure 1  Results of the PCA performed on morphometric data of the humerus.** (A) Distribution of the specimens along the two first axes of the PCA; (B–I) theoretical shapes associated with the minimum and maximum values of PC1: caudal (B, C), lateral (D, E), proximal (F, G) and distal (H, I) views for PC1 minimum (B, D, F, H) and PC1 maximum (C, E, G, I). B.g., bicipital groove; C., capitulum; D.t., deltoid tuberosity; E.c., epicondylar crest; G.t., greater tubercle; G.t.c., greater tubercle convexity; H., head; I.t., intermediate tubercle; L.e., lateral epicondyle; L.l.b., lateral lip border; L.t., lesser tubercle; L.t.c., lesser tubercle convexity; M.e., medial epicondyle; *M.i.*i., *M. infraspinatus* insertion; M.l.b., medial lip border; *M.t.m.*t., *M. teres major* tuberosity; N., neck; O.f., olecranon fossa; T., trochlea; T.g., trochlear groove.

tuberosity; a very large distal epiphysis because of the development of the lateral epicondyle; a smooth epicondylar crest; a mediolaterally wide and craniocaudally compressed medial epicondyle; shallow and proximodistally compressed olecranon fossa and trochlea; a wide trochlea displaying a main axis tilted in the dorsoventral direction; and a capitulum with a relatively small surface area. At the opposite, the theoretical shape at the PC1 maximum (Figs. 1C, 1E, 1G and 1I) shows a slender and thin aspect; a more rounded humeral head overhanging the diaphysis caudally; a greater tubercle more strongly developed than the lesser one and extending medially, conferring a more closed aspect to the bicipital groove, where the intermediate tubercle is almost absent; a poorly developed lesser tubercle convexity whereas the greater tubercle one is massive; a rounded and reduced *m. infraspinatus* insertion; a deltoid tuberosity strongly protruding laterally; a straight and thin shaft; no visible *m. teres major* tuberosity; a narrow distal epiphysis, with a small development of the lateral epicondyle; a sharp epicondylar crest; a craniocaudally developed medial epicondyle overhanging the olecranon fossa; a deep and wide olecranon fossa; a far less compressed trochlea, with a less dorsoventrally tilted axis; and a very reduced capitulum.

Along the second axis (7.6%), we observe this time that *C. simum* and *Dicerorhinus sumatrensis* are grouped together on the negative part of the axis, with the three other species on the positive part, whereas they are opposed along the first axis. This second axis expresses the separation between the lightest and the heaviest rhino species on the one hand and the three other species on the other hand. The theoretical shape at the PC2 minimum displays a humeral head stretched in the caudal direction; a lesser tubercle more developed than the greater one, delimiting an open bicipital groove; a proximodistally extended distal epiphysis, with an epicondylar crest starting almost on the middle of the shaft; a rounded and wide olecranon fossa. At the opposite, the theoretical shape at the PC2 maximum shows a rounded humeral head; a strong development of both tubercles and a more closed bicipital groove; a mediolaterally stretched distal epiphysis, with the epicondylar crest starting at the distal third of the shaft; an olecranon fossa proximodistally compressed and more rectangular; and a well-developed lateral epicondyle.

### Radius

The first two axes of the PCA performed on the radius express 52.3% of the total variance (Fig. 2A). The first axis (36.4%) opposes *Dicerorhinus sumatrensis* and *Diceros bicornis* to *R. unicornis* and *C. simum*. *R. sondaicus* overlaps both *R. unicornis* and *Diceros bicornis* clusters. The specimens of *Dicerorhinus sumatrensis* are split in two discrete clusters along the first axis, but no clear explanation linked to age, sex or geographic origin was associated with this distribution. Point dispersion along this axis indicates an important intraspecific variation for *Dicerorhinus sumatrensis*, and to a lesser extent for *Diceros bicornis* and *R. sondaicus*. Unlike for the humerus, phylogenetically related species are not grouped together on PCA and Neighbor Joining trees (Fig. S4B). The theoretical shape at the PC1 minimum (Figs. 2B, 2D, 2F and 2H) shows a massive morphology with large shaft and epiphyses; an asymmetrical proximal articular surface (constituting the ulnar notch), with a medial portion appearing nearly twice as large as the lateral one; a protruding lateral

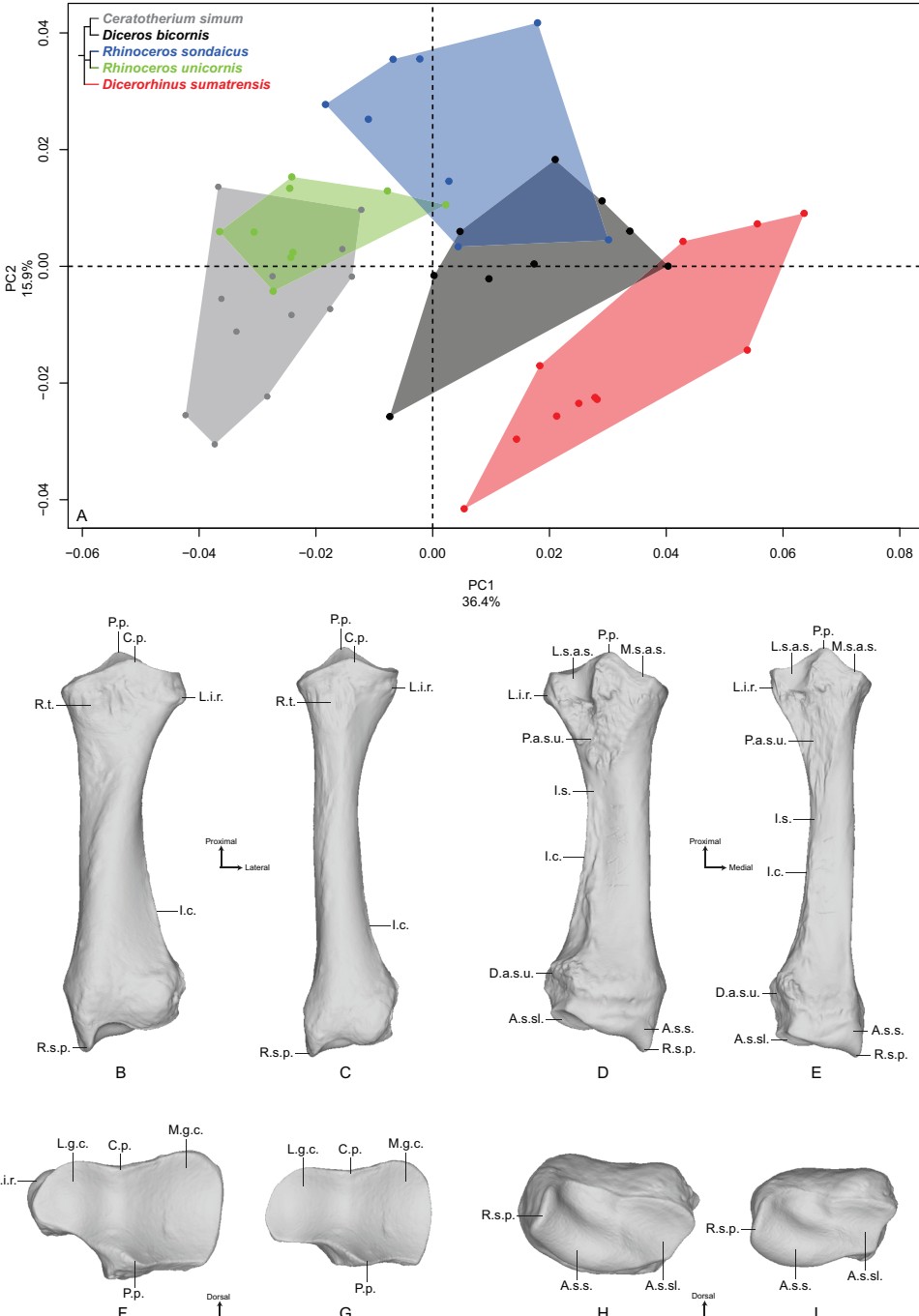

**Figure 2 Results of the PCA performed on morphometric data of the radius.** (A) Distribution of the specimens along the two first axes of the PCA; (B–I) theoretical shapes associated with the minimum and maximum values of PC1: dorsal (B, C), palmar (D, E), proximal (F, G) and distal (H, I) views for PC1 minimum (B, D, F, H) and PC1 maximum (C, E, G, I). A.s.s., articular surface for the scaphoid; A.s.sl., articular surface for the semilunar; C.p., coronoid process; D.a.s.u., distal articular surface for the ulna; I.c., interosseous crest; I.s., interosseous space; L.g.c., lateral glenoid cavity; L.i.r., lateral insertion relief; L.s.a.s., lateral synovial articular surface; M.g.c., medial glenoid cavity; M.s.a.s., medial synovial articular surface; P.a.s.u., proximal articular surface for the ulna; P.p., palmar process; R.s.p., radial styloid process; R.t., radial tuberosity.

insertion relief (distally to the lateral coronoid process sensu *Budras, Sack & Röck, 2009*) (i.e., insertion area of the *m. extensor digitorum communis*) whereas the radial tuberosity is little prominent; a mediolaterally reduced lateral synovial articulation surface for the ulna; a rectangular and thin medial synovial articulation surface for the ulna; a triangular proximal articular surface for the ulna as wide mediolaterally as proximodistally; a thick shaft with an interosseous space opening close to the proximal epiphysis: consequently, the interosseous crest runs along the diaphysis to the distal articular surface for the ulna; a broad distal epiphysis in the mediolateral direction, with a strong medial tubercle developed on the dorsal face; a distal articular surface compressed in the dorsoventral direction; an articular surface for the scaphoid little extended proximally; a trapezoidal and wide articular surface for the semilunar (i.e., lunate bone or lunatum); a well-developed radial styloid process. The theoretical shape at the PC1 maximum (Figs. 2C, 2E, 2G and 2I) displays a more slender morphology; a less asymmetrical proximal articular surface, despite the development of the medial part; an almost absent lateral insertion relief; a completely flat radial tuberosity; a mediolaterally reduced lateral synovial articulation surface for the ulna; a rectangular and thin medial synovial articulation for the ulna; a triangular proximal articular surface for the ulna, mediolaterally short and proximodistally stretched; a thin and slender shaft, with an interosseous space opening at the proximal third of the total length; a poorly visible interosseous crest; a less dorsoventrally compressed distal epiphysis and a poorly developed lateral tubercle (on the dorsal side); a dorsoventrally wide distal articular surface, with a proximally extended articular surface corresponding to the scaphoid; a trapezoidal and reduced articular surface for the semilunar; a less developed radial styloid process with a rounded border.

The second axis (15.9%) discriminates mainly *R. sondaicus* from the four other species. *R. unicornis* displays little extension along this axis; neither does *Diceros bicornis*, only driven on the negative side by a single individual. Extension of *R. unicornis* morphospace occupation along the second axis is very limited, contrary to that of *C. simum* and *Dicerorhinus sumatrensis* clusters. As on the first axis, *Dicerorhinus sumatrensis* is split in two clusters, one in the negative part and the other around null values. The theoretical shape at the PC2 minimum displays a slender morphology, with a strongly asymmetrical proximal articular surface; a proximally reduced palmar process, opposed to the coronoid process; a distal epiphysis dorsoventrally broad, with a developed lateral prominence; a little developed radial styloid process; a slight proximal extension of the articular surface for the scaphoid. The theoretical shape at PC2 maximum displays a more massive shape; a deeper and more symmetrical proximal articular surface with a well-developed palmar process; a dorsoventrally compressed distal epiphysis with a more developed styloid process.

### Ulna

The first two axes of the PCA performed on the ulna express 41.5% of the total variance (Fig. 3A). The first axis (22.1%) separates *Dicerorhinus sumatrensis* and *Diceros bicornis* on the positive part and *R. sondaicus*, *R. unicornis* and *C. simum* on the negative part. However, the clusters of *C. simum* and *R. unicornis* overlap along this axis. The general

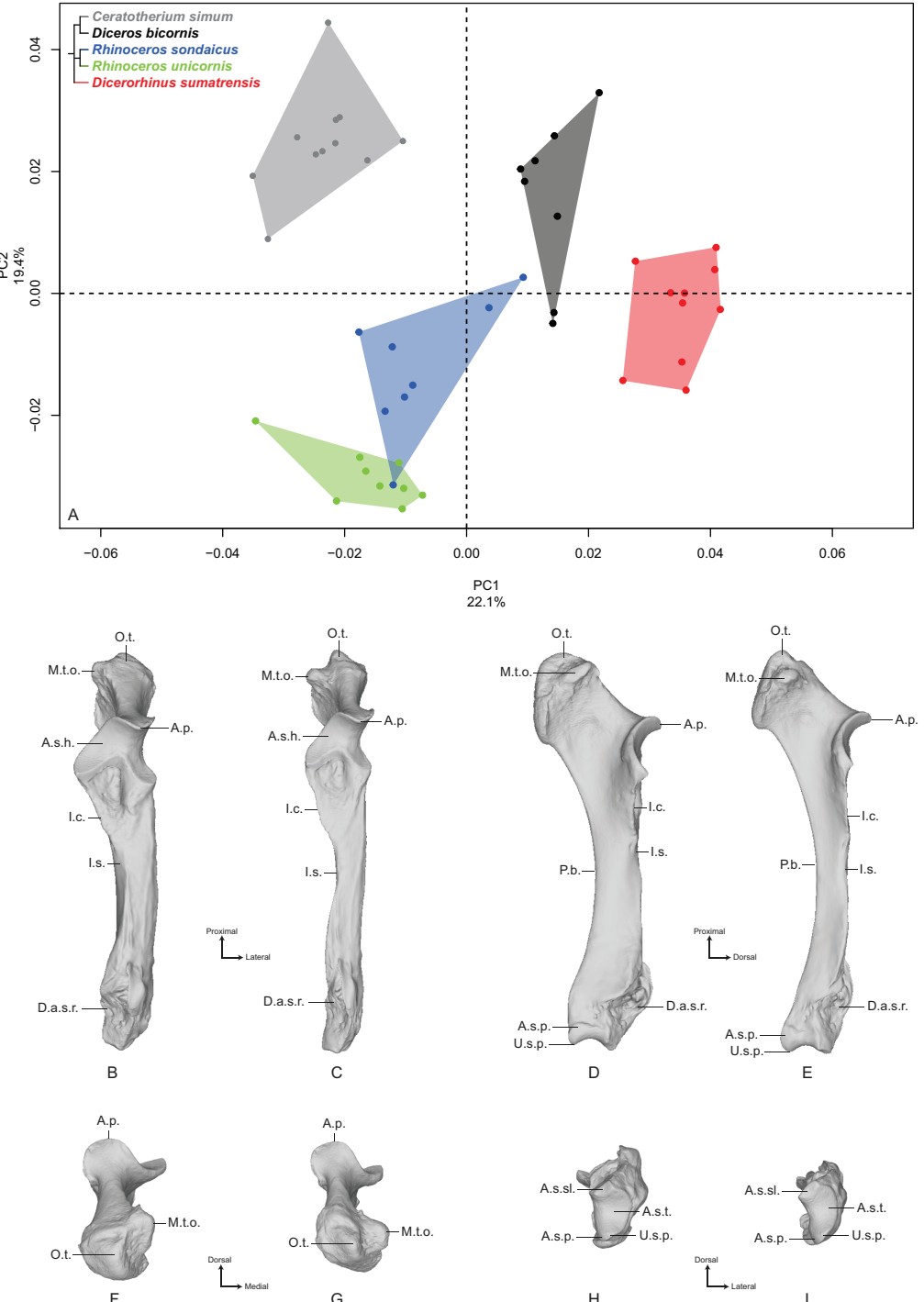

**Figure 3 Results of the PCA performed on morphometric data of the ulna.** (A) Distribution of the specimens along the two first axes of the PCA; (B–I) theoretical shapes associated with the minimum and maximum values of PC1: dorsal (B, C), medial (D, E), proximal (F, G) and distal (H, I) views for PC1 minimum (B, D, F, H) and PC1 maximum (C, E, G, I). A.p., anconeal process; A.s.h., articular surface for the humerus; A.s.p., articular surface for the pisiform; A.s.sl., articular surface for the semilunar; A.s.t., articular surface for the triquetrum; D.a.s.r., distal articular surface for the radius; I.c., interosseous crest; I.s., interosseous space; M.t.o., medial tuberosity of the olecranon; O.t., olecranon tuberosity; P.b., palmar border; U.s.p., ulnar styloid process.

pattern on both PCA and Neighbor Joining trees is close to the one observed for the radius (Fig. S4C). The theoretical shape at the PC1 minimum (Figs. 3B, 3D, 3F and 3H) displays a thick morphology with large epiphyses; a massive olecranon tuberosity with a medial tubercle—where inserts the medial head of the *m. triceps brachii*, as well as the *mm. flexor carpi ulnaris* and *flexor digitorum superficialis*—oriented dorsally; an anconeal process poorly developed dorsally and mediolaterally wide, as is the articular surface constituting the trochlear notch (receiving the humeral trochlea); a medially stretched medial part of the articular surface for the humerus; a short interosseous crest ending at the shaft half, with the interosseous space; a broad shaft with a triangular cross-section; a straight palmar border whereas the shaft is medially curved; a massive distal epiphysis with a wide insertion surface for the radius; a mediolaterally wide and little concave articular surface for the triquetrum (i.e., triquetral, pyramidal or cuneiform bone), while the one responding to the pisiform is crescent-shaped and little extended proximally. The theoretical shape for the PC1 maximum (Figs. 3C, 3E, 3G and 3I) displays a more gracile morphology; a slender olecranon tuberosity with a medial tubercle where inserts the medial head of the *m. triceps brachii* oriented in the palmar direction; a dorsally developed and mediolaterally narrow anconeal process, as is the articular surface of the trochlear notch; a slightly medially stretched medial part of the articular surface; a sharp interosseous crest; a thin and straight shaft; a mediolaterally compressed and little concave distal epiphysis; a mediolaterally narrow articular surface for the triquetrum; a triangular and proximally well-developed articular surface for the pisiform.

The second axis (19.4%) separates quite clearly the three Asian species from the African ones. The theoretical shape at the PC2 minimum displays a slender and straight morphology with a high square-shaped olecranon process, mediolaterally flattened, more stretched in the palmar direction; a wide and squared anconeal process; a straight and regular shaft; a mediolaterally compressed distal epiphysis with a concave articular surface for the triquetrum and a distally developed styloid process; a proximally extended articular facet for the pisiform. The theoretical shape at the PC2 maximum displays a more massive and medially concave shape with an olecranon process mediolaterally inflated and rounded in the palmar direction; an anconeal process little developed dorsally and laterally tilted; a proximodistally compressed and extending medially articular surface constituting the trochlear notch; a mediolaterally wide articular surface for the triquetrum; a little developed styloid process; a poorly extended proximally and square-shaped articular surface for the pisiform.

### Femur

The first two axes of the PCA performed on the femur express 45.0% of the global variance (Fig. 4A). The first principal component (36.1%) clearly isolates *Dicerorhinus sumatrensis* on the positive part from the other species. The clusters of *Diceros bicornis*, *R. sondaicus* and *R. unicornis* overlap on the negative part of the axis. *Diceros bicornis* and *R. unicornis* specimens overlap a substantial part of the cluster of *C. simum* too. The general pattern observed on the Neighbor Joining tree is closer to the humerus one, with African and Asiatic species grouped together, respectively (Fig. S4D). The theoretical

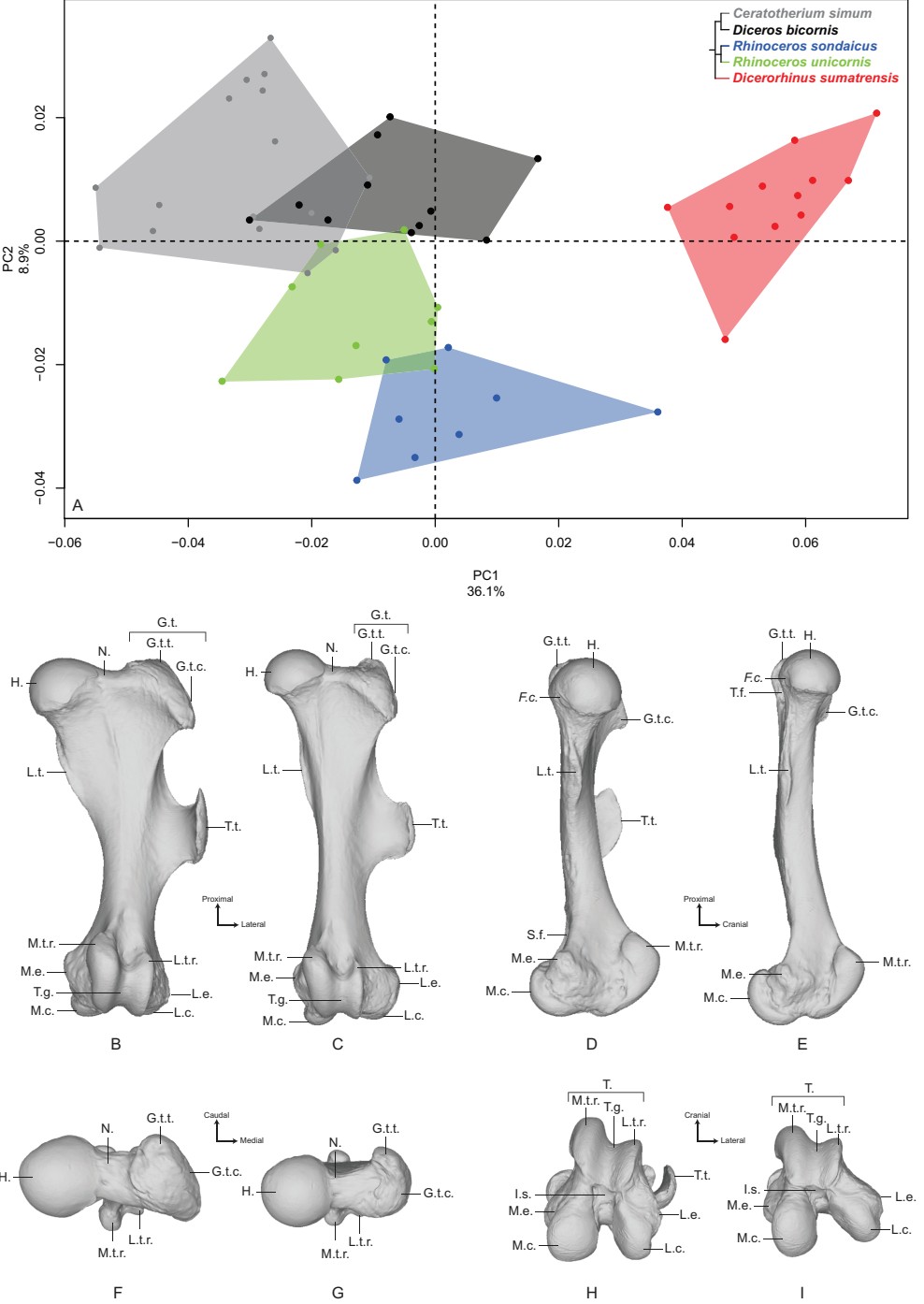

**Figure 4 Results of the PCA performed on morphometric data of the femur.** (A) Distribution of the specimens along the two first axes of the PCA; (B–I) theoretical shapes associated with the minimum and maximum values of PC1: cranial (B, C), medial (D, E), proximal (F, G) and distal (H, I) views for PC1 minimum (B, D, F, H) and PC1 maximum (C, E, G, I). *F.c.*, *Fovea capitis*; G.t., greater trochanter; G.t.c., greater trochanter convexity; G.t.t., greater trochanter top; H., head; I.s., intercondylar space; L.c., lateral condyle; L.e., lateral epicondyle; L.t.r., lateral trochlear ridge; L.t., lesser trochanter; M.c., medial condyle; M.e., medial epicondyle; M.t.r., medial trochlear ridge; N., neck; S.f., supracondylar fossa; T., trochlea; T.f., trochanteric fossa; T.g., trochlear groove; T.t., third trochanter.

shape at the PC1 minimum (Figs. 4B, 4D, 4F and 4H) shows a massive morphology with large epiphyses and a curved medial border, conferring a concave aspect to the diaphysis; a large femoral head, off-centered relatively to the shaft main axis, supported by a very large neck; a small and shallow *fovea capitis* mediocaudally oriented; a greater trochanter convexity expanding strongly laterodistally; the absence of trochanteric notch between the convexity and the top of the trochanter (Fig. 4F); a proximodistally reduced trochanteric fossa; a sharp lesser trochanter running along the medial edge, which is craniocaudally flattened below the humeral head; a third trochanter extending strongly laterally, cranially and proximally toward the greater trochanter convexity, and much curved toward the medial direction; a quite irregular shaft section along the bone—flattened below the proximal epiphysis and more trapezoidal toward the distal epiphysis; a broad distal epiphysis with developed medial and lateral epicondyles; a shallow supracondylar fossa; a wide trochlea, with a main rotation axis aligned with the shaft axis; a large and cranially expanded medial ridge of the trochlea separated from the lateral one by a deep trochlear groove; a medial condyle surface area larger than the lateral condyle one, both being separated by a narrow intercondylar space. At the opposite, the theoretical shape at the PC1 maximum (Figs. 4C, 4E, 4G and 4I) is more slender with a straight and regular shaft; a rounded femoral head aligned with the shaft main axis and supported by a thinner neck; a more pronounced and rounded *fovea capitis* oriented almost completely caudally; a greater trochanter convexity little developed laterodistally; a more pronounced trochanter top despite the absence of trochanteric notch; a thin lesser trochanter situated on the caudal border of the medial side; a rounded third trochanter more developed laterally than cranially; a quite regular and trapezoidal shaft section; a mediolaterally broader and medially oriented distal epiphysis; an almost absent supracondylar fossa; a less developed medial trochlear ridge separated from the lateral one by a shallow trochlear groove; a lateral condyle more oblique and divergent relatively to the medial one, increasing the intercondylar space; symmetrical medial and lateral condylar surfaces.

The second axis (8.9%) clearly opposes *Dicerorhinus sumatrensis*, *C. simum* and *Diceros bicornis* on the positive part to the two *Rhinoceros* species on the negative part, the cluster of *Dicerorhinus sumatrensis* being driven toward negative values by a single individual. The theoretical shape at the PC2 minimum is mainly characterized by a flattened femoral head with a strong neck; a rounded and large *fovea capitis* mediocaudally oriented; a laterodistally expanded greater trochanter convexity; a long and thin lesser trochanter; an extremely developed third trochanter in lateral, cranial and proximal directions; a straight and regular shaft; a broad distal epiphysis with important development of both epicondyles; a trochlea rotation axis aligned with the main axis of the shaft. The theoretical shape at the PC2 maximum displays a more rounded head, with a more stretched neck; no *fovea capitis* at all but a little groove on the head border; a greater trochanter convexity little expanded laterodistally; a short and more medially developed lesser trochanter; a rounded third trochanter little developed in cranial and proximal directions; a straight shaft; a distal epiphysis less mediolaterally broad; a narrower intercondylar space; a more inflated medial condyle.

### Tibia

The first two axes of the PCA performed on the tibia express 50.0% of the global variance (Fig. 5A). The first axis (29.1%) separates roughly *Diceros bicornis* and *Dicerorhinus sumatrensis* on the positive part and *C. simum*, *R. sondaicus* and *R. unicornis* on the negative part. *Diceros bicornis* shows an important intraspecific variation along both axes. Neighbor Joining tree structure is less clear than for previous bones: both *Rhinoceros* species isolate from most of the other specimens, *C. simum* appears also separated from *Diceros bicornis* and *Dicerorhinus sumatrensis*. However, one *C. simum* and three *Dicerorhinus sumatrensis* specimens are closer from the *Rhinoceros* group than from their own respective species (Fig. S4E). The theoretical shape at the PC1 minimum (Figs. 5B, 5D, 5F and 5H) shows a massive morphology with broad shaft and epiphyses, both in craniocaudal and mediolateral directions; medial and lateral intercondylar tubercles having the same height and a reduced central intercondylar area; a broad cranial intercondylar area; a medial articular surface larger than the lateral one, with a caudally extended sliding surface for the *m. popliteus* tendon; a U-shaped popliteal notch; a rounded tibial tuberosity, laterally deflected and medially bordered by a shallow groove; a shallow extensor groove; a regularly triangular and distally extended proximal articular surface for the fibula; a thick tibial crest disappearing at the middle of the shaft, where the bone section is the smallest; a mediolaterally broad and rectangular in section distal epiphysis; a triangular-shaped distal articular surface for the fibula reduced in height, surmounted by a smooth interosseous crest running toward the middle of the shaft; a roughly rectangular distal articular surface for the talus, with a lateral groove larger and shallower than the medial one, separated by a prominent intermediate process without synovial fossa; an articular surface with a rotation axis aligned with the bone main axis; a prominent medial malleolus. The theoretical shape at the PC1 maximum (Figs. 5C, 5E, 5G and 5I) displays a relatively gracile morphology with a thin shaft; a lateral intercondylar tubercle more proximally extended than the medial one and a relatively large central intercondylar area; a cranially extended lateral condylar surface, reducing the cranial intercondylar area; roughly equal medial and lateral articular surface areas; a V-shaped popliteal notch; a tibial tuberosity slightly more laterally deflected; a deeper tuberosity groove; a nail-shaped proximal articular surface for the fibula; a sharper tibial crest disappearing just before the first half of the shaft; a distal epiphysis more compressed craniocaudally; a distal articular surface for the fibula displaying a large triangle synostosis area occupying a third of the shaft and prolonged by a sharp interosseous crest. There is no major difference in the distal articular shape between PC1 maximum and minimum, except that the caudal apophysis is less prominent in the distal direction.

The second axis (20.9%) clearly separates the two African species (*C. simum* and *Diceros bicornis*) on the positive part from the three Asian species (*Dicerorhinus sumatrensis*, *R. sondaicus* and *R. unicornis*) on the negative part. The theoretical shape at the PC2 minimum displays a slightly more slender morphology; a proximal plateau higher cranially than caudally and forming a closer angle with the diaphysis axis; a high intercondylar eminence; a lateral articular surface more caudally extended than the medial one; a tibial

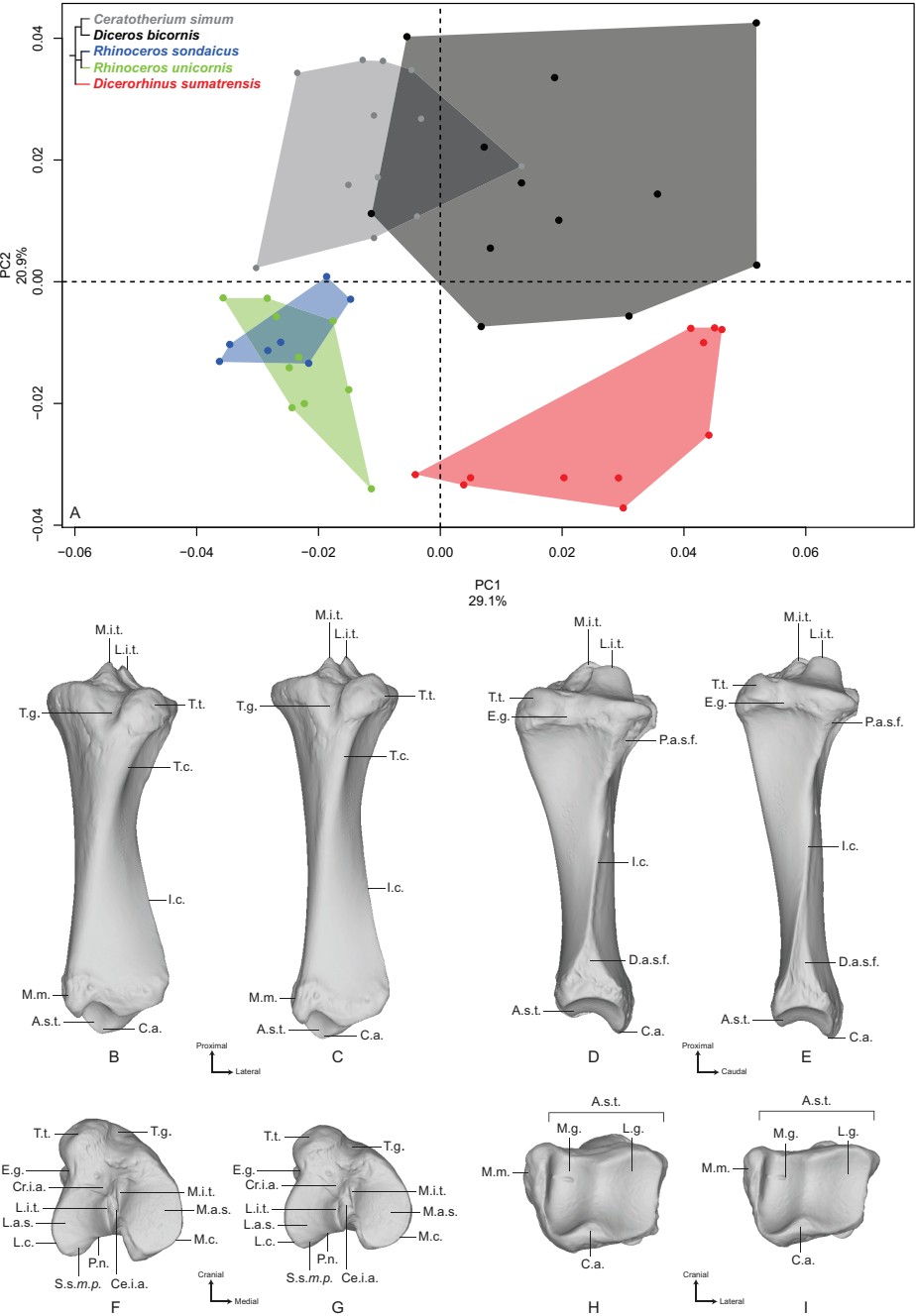

Peer

**Figure 5 Results of the PCA performed on morphometric data of the tibia.** (A) Distribution of the specimens along the two first axes of the PCA; (B–I) theoretical shapes associated with the minimum and maximum values of PC1: cranial (B, C), lateral (D, E), proximal (F, G) and distal (H, I) views for PC1 minimum (B, D, F, H) and PC1 maximum (C, E, G, I). A.s.t., articular surface for the talus; C.a., caudal apophysis; Ce.i.a., central intercondylar area; Cr.i.a., cranial intercondylar area; D.a.s.f., distal articular surface for the fibula; E.g., extensor groove; I.c., interosseous crest; L.a.s., lateral articular surface; L.c., lateral condyle; L.g., lateral groove; L.i.t., lateral intercondylar tubercle; M.a.s., medial articular surface; M.c., medial condyle; M.g., medial groove; M.i.t., medial intercondylar tubercle; M.m., medial malleolus; P.a.s.f., proximal articular surface for the fibula; P.n., popliteal notch; S.s.*m.p.*, sliding surface for the *m. popliteus*; T.c., tibial crest; T.g., tuberosity groove; T.t., tibial tuberosity.

tuberosity well separated from the condyles by deep tuberosity and extensor grooves; a straight shaft ending with divergent borders forming a large and rectangular distal epiphysis; a distal articular surface for the fibula forming a regular triangle surmounted by a sharp interosseous crest; a medially extended medial malleolus, resulting in a rectangular articular surface with the talus, where the medial groove is narrow and deep, occupying a third of the area, whereas the lateral groove is shallow and broad. The theoretical shape at the PC2 maximum displays a more massive morphology, with a craniocaudal inflation of the epiphyses; a proximal plateau almost perpendicular to the diaphysis axis; a lower intercondylar eminence; a lateral condyle surface almost twice less large than the medial one, which is more developed caudally; a massive tibial tuberosity strongly deviated laterally, delimited by very shallow tuberosity and extensor grooves and resulting in a very large cranial intercondylar area; a straight shaft ending with almost parallel medial and lateral borders and a square-shaped distal epiphysis; a less medially deflated medial malleolus; a squared distal articular surface for the talus with medial and lateral grooves showing similar surface area and depth.

### Fibula

The first two axes of the PCA performed on the fibula express 55.9% of the global variance (Fig. 6). Contrary to the five previous analyses, the first axis (40.7%) here seems particularly driven by a strong intraspecific variation. The clusters of *C. simum* and *Dicerorhinus sumatrensis* are stretched along the PC1 and overlap with almost every other specimens. The cluster of *Diceros bicornis* is quite stretched along the axis too and only the two *Rhinoceros* species display less intraspecific variation. This pattern does not seem linked to sex, age class or condition (wild or captive): despite the presence of slightly more females and subadults on the negative part of the component, we did not consider this observation as robust enough to state on this question. This cluster distribution along the PC1 seems linked to the presence of irregular crests along the shaft, associated with an important variation of the outline of the crests running along the diaphysis, and a slight rotation of the fibular head (see Fig. S3). Consequently, we chose to display and analyze the specimen distribution along the second and third components instead. Theoretical shapes associated with the PC1 are available in Fig. S3.

PC2 and PC3 express 22.9% of the global variance (Fig. 7A). The second component (15.2%) opposes *C. simum* on the negative side to *Dicerorhinus sumatrensis* on the positive side, whereas *Diceros bicornis*, *R. sondaicus* and *R. unicornis* have a more central disposition. As for the tibia, the Neighbor Joining tree structure appears less clearly sorted by species than for other bones. If *Rhinoceros* species group together and African ones as well, *Dicerorhinus sumatrensis* sample is split in two subgroups mixed with *R. unicornis* and African rhinos, respectively (Fig. S4F). The theoretical shape at the PC2 minimum (Figs. 7B, 7D, 7F, 7H and 7J) displays a broad morphology with large epiphyses and a straight shaft; a rounded head with a craniomedially oriented proximal articular surface for the tibia; a head width similar to the shaft one; a robust shaft with two strong craniolateral and caudolateral lines running down the distal epiphysis and enlarging craniocaudally toward the distal epiphysis; a sharp and irregular interosseous crest; a mediolaterally

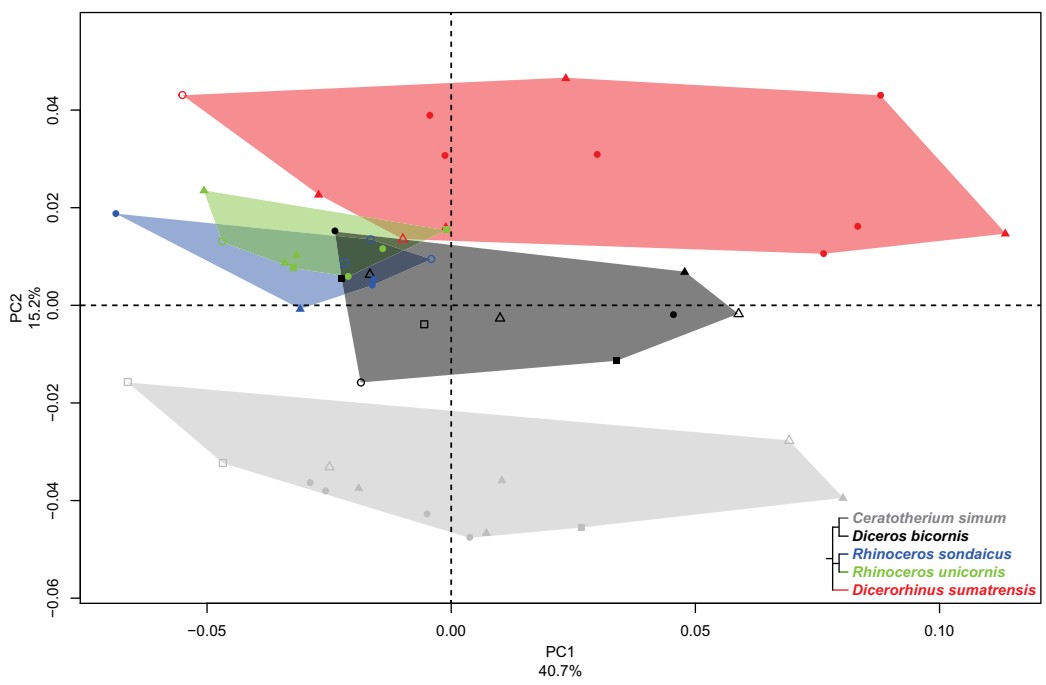

**Figure 6 Results of the PCA performed on morphometric data of the fibula.** Distribution of the specimens along the two first axes of the PCA, taking into account the age class and the sex of each specimen. Square, female; triangle, male; circle, unknown; empty symbol, subadult; filled symbol, adult.

compressed distal epiphysis with little development of the two distal tubercles at the end of the lateral crests; a shallow lateral groove; a triangular distal articular surface for the tibia, occupying only the last distal quarter of the bone length; a short and ovoid articular surface for the talus with a sharp distal ridge. The theoretical shape at the PC2 maximum (Figs. 7C, 7E, 7G, 7I and 7K) displays a slender morphology with a strongly curved shaft; a mediolaterally flat head extending craniocaudally and overhanging strongly the diaphysis; a thin shaft with two sharp lateral crests running along it: these crests end with two developed tubercles surrounding a deep lateral groove; a distal articular surface for the tibia extending from the distal third of the shape and forming a stretched triangle; a wider and kidney-shaped articular surface for the talus, forming two distal tips corresponding to the two lateral tubercles: between them on the distal face, a large groove is visible, ending at the center of the face.

The third component (7.7%) mainly opposes *Diceros bicornis* on the positive part to *R. sondaicus* on the negative part. However, this opposition is mainly driven by a small number of individuals (two for *Diceros bicornis* and four for *R. sondaicus*). The specimens of *R. sondaicus* are divided into two clusters, with three individuals overlapping notably with *Dicerorhinus sumatrensis*. The theoretical shape at the PC3 minimum shows a massive morphology, with broad shaft and epiphyses; a craniocaudally broad head, overhanging the shaft laterally; a proximal articular surface for the tibia almost completely medially oriented; a straight shaft displaying a constant width along the bone; craniolateral and caudolateral crests running almost parallel toward the distal end of the bone, forming

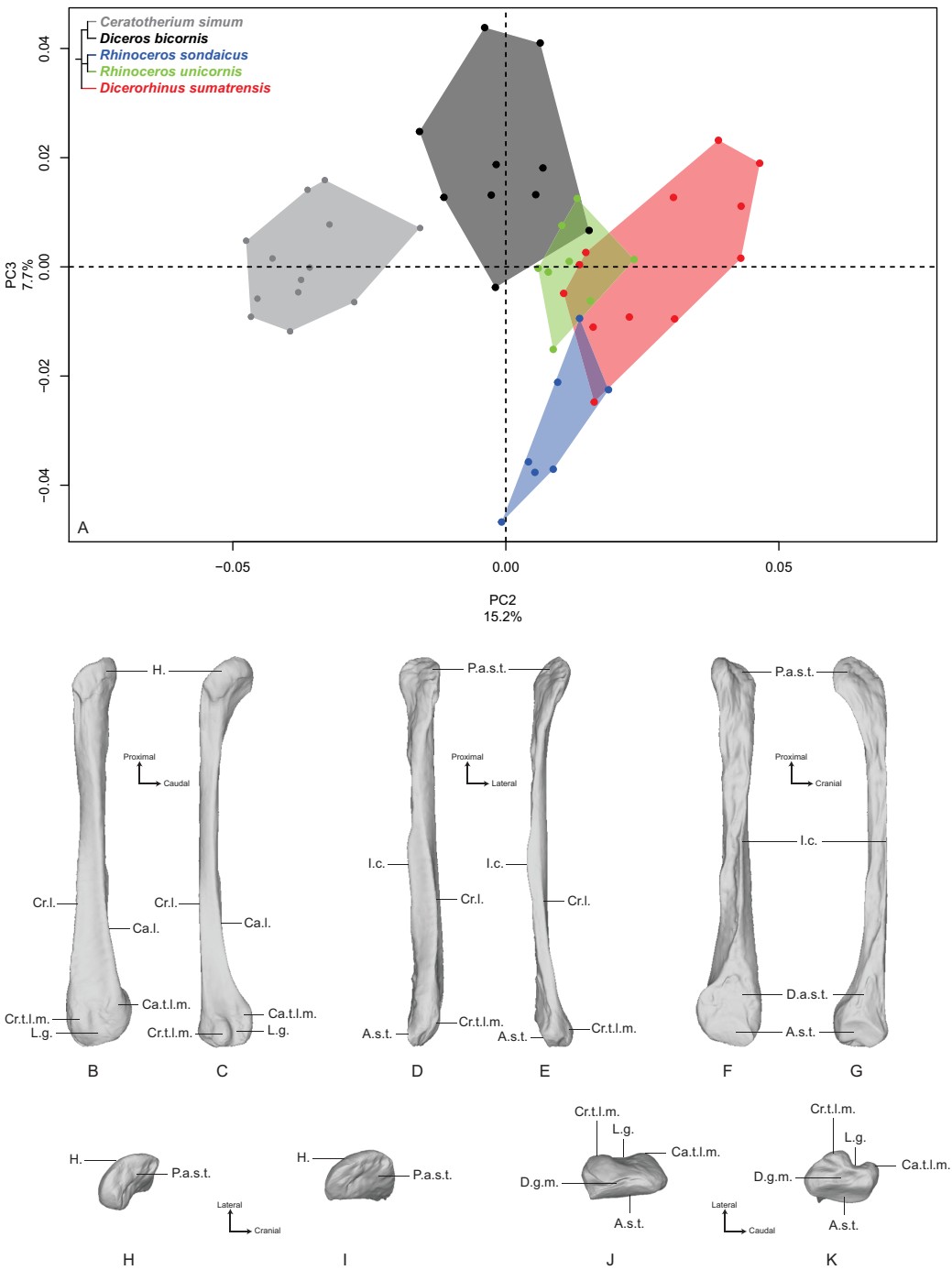

**Figure 7 Results of the PCA performed on morphometric data of the fibula (second and third axes).**
(A) Distribution of the specimens along the second and third axes of the PCA; (B–K) theoretical shapes associated with the minimum and maximum values of PC2: lateral (B, C), cranial (D, E), medial (F, G), proximal (H, I) and distal (J, K) views for PC2 minimum (B, D, F, H, J) and PC2 maximum (C, E, G, I, K). A.s.t., articular surface for the talus; Ca.l., caudo-lateral line; Ca.t.l.m., caudal tubercle of the lateral malleolus; Cr.l., cranio-lateral line; Cr.t.l.m., cranial tubercle of the lateral malleolus; D.a.s.t., distal articular surface for the tibia; D.g.m., distal groove of the malleolus; H., head; I.c., interosseous crest; L.g., lateral groove; P.a.s.t., proximal articular surface for the tibia.

two developed tubercles surrounding a deep groove; an interosseous space covered by irregular reliefs and bordered by a sharp interosseous crest; a distal articular surface for the tibia forming a cranially deviated triangle; a kidney-shaped distal articular surface for the talus, with a distal border separated from the lateral tubercles by a groove stopping at the middle of the distal face. The theoretical shape at the PC3 maximum shows an extremely thin morphology with a flattened and poorly developed head; a proximal articular surface oriented almost completely in the cranial direction; a torsion of almost 90° between the orientation of the proximal and distal articular surfaces for the tibia; a very thin and flat shaft; craniolateral and caudolateral crests running along the diaphysis ending on the distal epiphysis with few developed tubercles; a distal articular surface for the tibia forming a slender triangle; a relatively small distal articular surface for the talus, with a less pronounced kidney-shape; a groove on the distal face mediolaterally compressed.

## Interspecific morphological variation

In addition to global interspecific patterns of shape, we shortly describe the main morphological features characterizing each species. Mean shapes of each bone for the five species are available in Fig. S5.

Limb long bones of *C. simum* present a general massive and robust aspect. The humerus is thick and shows a strong development of the lesser tubercle and the lateral epicondyle, as well as a proximal broadening in the craniocaudal direction. The radius and ulna are robust and display an important medial development of the articular parts constituting the trochlear notch. The ulna bears a strong olecranon tubercle. The distal articular surface for the carpals constituted by the two bones is mediolaterally wide and compressed in the craniocaudal direction. The hind limb bones are robust as well, this robustness being mainly expressed in the mediolateral direction for the femur. This bone displays a rounded and thick head, strong greater and third trochanters, and a distal trochlea laterally oriented. The tibia and fibula are robust as well, with a wide tibial plateau supporting the knee articulation and a squared distal articulation for the talus.

For *Diceros bicornis*, the general aspect of the humerus is close to the one observed on *C. simum*, particularly for the epiphyses (e.g., the shape of the bicipital groove, the development of the lesser tubercle and of the lateral epicondyle), though its degree of robustness is less intense. The radius is relatively slender but the proximal articular surface displays a cranial border with a marked groove under the coronoid process, also observed on *C. simum*. The ulna is slender as well with a thin olecranon process and limited medial development. Both distal epiphyses form a mediolaterally wide articular surface for the carpals, poorly craniocaudally compressed. As for hind limb bones, the femur is only slightly robust, with poorly developed trochanters and a slender diaphysis. Tibia and fibula are less thick too, with a squared articular surface for the talus as well. *Diceros bicornis* displays noticeable morphological similarities with *C. simum*.

The bone general morphology is very similar between both *R. sondaicus* and *R. unicornis*, being often more robust in *R. sondaicus*. For these two species, the humerus displays an important development of both lesser and greater tubercles, resulting in an asymmetrical bicipital groove. The greater tubercle is even sometimes higher than the

lesser one in *R. sondaicus*, which is not the case in *R. unicornis*. The distal epiphysis is wide but with a medial epicondyle less developed than in *C. simum* and *Diceros bicornis*, and a rectangular olecranon fossa. The radius exhibits mediolaterally large epiphyses and a quite robust diaphysis, with a proximal articular surface similar in both *Rhinoceros* species, with a straight cranial border unlike in African rhinos. The distal epiphysis is rectangular and craniocaudally compressed. *R. unicornis* distinguishes from *R. sondaicus* in having a more robust radius, with a more asymmetrical proximal epiphysis, a deeper radial tuberosity and a larger distal articular surface. The ulna is also very similar, the one of *R. unicornis* being slightly more robust. The general aspect remains extremely close, with a developed olecranon, a medial development of the articular surface constituting the trochlear notch and a quite wide distal articular surface. On the hind limb, the femur appears different, the *R. unicornis* one showing important development of the greater and third trochanters, sometimes fused by a bony bridge as previously stated by *Guérin (1980)*. The femur of *R. sondaicus* appears slightly less robust, and the greater and third trochanters are less developed and never fused. On the tibia, the proximal plateau is as wide as for the African taxa but the tibial tuberosity is more detached from the condyles by deep tuberosity and extensor grooves. The diaphysis is relatively thick and the distal articular surface is clearly rectangular. The fibula is very similar as well in the two species, with a distal epiphysis curved in the caudal direction and a kidney-shaped articular surface for the talus.

*Dicerorhinus sumatrensis* clearly differs from the other species. Despite clear rhinocerotid features, limb long bones display unique morphological traits, with a more pronounced slenderness. On the humerus, the development of the greater tubercle results in a more closed and asymmetrical bicipital groove. The distal epiphysis is mediolaterally narrow with a straight trochlea axis. The thin radius possesses a proximal articular surface almost symmetrical despite a medial glenoid cavity slightly more developed. The ulna is thin as well, and forms with the radius a rectangular articular surface for the carpals. The femur shows a high and rounded head and a poorly developed third trochanter. The distal trochlea axis is more medially oriented. On the tibia, the plateau is far less wide than in other species and the distal articular surface for the talus is rectangular. The thin fibula displays a large head caudally bordered by a thin crest and the diaphysis is strongly curved medially toward the tibia. The kidney-shape of the distal articular surface for the talus resembles the *Rhinoceros* ones.

## Correlation with the centroid size

Table 3 provides the results of the Pearson's correlation tests between the centroid size and the two first principal components for each bone (and the third component for the fibula). There is a significant correlation in each case between the first component and the centroid size, with higher correlation coefficient values for the radius and ulna, and smaller values for the humerus and fibula. The second principal component is also significantly correlated with the centroid size for the humerus, femur and fibula, with smaller correlation coefficient values than for PC1, except for the humerus.

**Table 3 Results of the Pearson's correlation tests between the log-transformed centroid size and the two first principal components for each bone.**

| Bone | Component | r | t | dF | P |
|------|-----------|-----|-----|-----|-----|
| Humerus | PC1 | −0.38 | −2.93 | 51 | **0.01** |
|  | PC2 | 0.43 | 3.44 | 51 | **<0.01** |
| Radius | PC1 | −0.64 | −5.77 | 47 | **<0.01** |
|  | PC2 | 0.22 | 1.58 | 47 | 0.12 |
| Ulna | PC1 | −0.79 | −8.44 | 44 | **<0.01** |
|  | PC2 | 0.02 | 0.11 | 44 | 0.91 |
| Femur | PC1 | −0.56 | −5.01 | 54 | **<0.01** |
|  | PC2 | 0.30 | −2.34 | 54 | **0.02** |
| Tibia | PC1 | −0.58 | −5.05 | 51 | **<0.01** |
|  | PC2 | 0.08 | 0.58 | 51 | 0.57 |
| Fibula | PC1 | −0.36 | −2.69 | 48 | **<0.01** |
|  | PC2 | −0.34 | −2.47 | 48 | **0.02** |
|  | PC3 | 0.16 | 1.12 | 48 | 0.27 |

Notes:
  *r*, Pearson's correlation coefficient value; *t*, student distribution value; dF, degrees of freedom; *P*, *p*-value.
  Significant results are indicated in bold.

## Allometry

Tables 4 and 5 provide the main anatomical differences observed between theoretical shapes associated with minimal and maximal centroid size for the forelimb and hind limb bones, respectively. Theoretical shapes associated with minimal and maximal log centroid size are provided in Fig. S6. In the case of the fibula, we found a pattern very close to the one observed along the second axis of the PCA. Replacing the log centroid size by the cube root of the mean mass of each species results in almost identical theoretical shapes for each bone (Fig. 8; Fig. S7), only distinguishable by minor shape differences: toward body mass maximum, the radius and ulna appear slightly more robust than for centroid size maximum (Figs. 8D and 8F); the greater and third trochanters of the femur are slightly less developed toward each other (Fig. 8H). Theoretical shapes associated with minimum and maximum of log centroid size are slightly more massive than the ones obtained with the cube root of the body mass for the humerus, the tibia and the fibula. All theoretical shapes associated with minimal and maximal cube root of the mean mass are provided in Fig. S7.

Tables 6 and 7 provide the results of the two Procrustes ANOVAs performed on shape data, where the log centroid size and the cube root of the mean body mass were, respectively, the independent variable. Log centroid size is significantly correlated with shape for the six bones, with a determination coefficient varying between 0.10 for the fibula and 0.18 for the ulna. In every case, the determination coefficient is more than twice as high for species affiliation as for log centroid size, indicating a more important influence of group affiliation than of allometry. This is especially the case for the humerus, with a determination coefficient of 0.53 for species affiliation and of only 0.13 for log centroid size. Cube root of mean body mass is also significantly correlated with shape for the six bones, with slightly higher determination coefficient values than those obtained with the

**Table 4 Main anatomical differences observed between theoretical shapes associated with minimal and maximal centroid size for each bone of the forelimb.**

| B | Anatomical feature | Centroid size minimum | Centroid size maximum |
|---|---|---|---|
| H | General aspect | Gracile | Robust |
| | Head | Rounded, overhanging the shaft | Rounded, overhanging poorly the shaft |
| | Lesser tubercle | Developed | Poorly developed |
| | Intermediate tubercle | Almost absent | Poorly developed |
| | Greater tubercle | Developed | Strongly developed |
| | Bicipital groove | Asymmetrical and closed | Almost symmetrical and widely open |
| | *M. infraspinatus* insertion | Diamond-shaped and strongly developed | Ovoid and less developed |
| | Deltoid tuberosity | Poorly laterally deviated and caudally sharp | Laterally deviated and caudally smooth |
| | Distal epiphysis | Medio-laterally compressed | Medio-laterally extended |
| | Supracondylar crest | Smooth | Very smooth |
| | Lateral epicondyle | Poorly extended laterally | Strongly extended laterally |
| | Medial epicondyle | Overhanging the olecranon fossa | Not overhanging the olecranon fossa |
| | Olecranon fossa | Triangular and deep | Rectangular and deep |
| | Trochlea | Sharp lips and deep groove | Smooth lips and shallow groove |
| | Capitulum | Extremely reduced | Extremely reduced |
| R | General aspect | Gracile | Robust |
| | Proximal articular surface | Open and little concave; medial glenoid cavity slightly larger than the lateral one | Concave; medial glenoid cavity twice as large as the lateral one |
| | Radial tuberosity | Poorly developed | Poorly developed |
| | Lateral insertion relief | Poorly developed | Knob-shaped |
| | Lateral synovial articulation surface | Trapezoid and laterally extended | Trapezoid and laterally reduced |
| | Medial synovial articulation surface | Thin and rectangular | Thin and rectangular |
| | Proximal articular surface for the ulna | Triangular, wide and proximo-distally short | Triangular, slender and proximo-distally long |
| | Interosseous crest | Smooth | Sharp |
| | Interosseous space position | Mid-shaft | First proximal third of the shaft |
| | Distal articular surface for the ulna | Long and slender triangle | Short and wide triangle |
| | Articular surface for the carpal bones | Broad in dorso-palmar direction | Compressed in dorso-palmar direction |
| | Articular surface for the scaphoid | Proximally extended | Poorly extended proximally |
| | Articular surface for the semilunar | Trapezoid and narrow | Trapezoid and wide |
| | Radial styloid process | Short | Long |
| U | General aspect | Gracile | Robust |
| | Olecranon | Medio-laterally compressed | Medio-laterally large |
| | Olecranon tuberosity | Oriented medially with a medial tubercle pointing in the medio-palmar direction | Oriented laterally with a medial tubercle pointing in the medio-dorsal direction |
| | Anconeus process | Developed in dorsal direction | Little developed dorsally |
| | Articular surface for the humerus | Medio-laterally reduced, lateral lip developed in proximal direction | Medio-laterally broad with an important development of the medial part |
| | Interosseous crest | Irregular and sharp | Smooth |
| | Distal epiphysis | Thin with a small lateral extension | Large and extending largely in lateral and dorsal directions |
| | Articular surface for the triquetrum | Narrow and concave | Wide and slightly concave |
| | Articular surface for the pisiform | Extended in proximal direction | Little developed in proximal direction |

**Note:**
B, bone; H, humerus; R, radius; U, ulna.

**Table 5 Main anatomical differences observed between theoretical shapes associated with minimal and maximal centroid size for each bone of the hind limb.**

| B | Anatomical feature | Centroid size minimum | Centroid size maximum |
|---|---|---|---|
| Fe | General aspect | Gracile | Robust |
| | Head | Rounded, well separated from the shaft by a narrow neck | Massive and flattened, surmounting a large neck |
| | *Fovea capitis* | Formed by a simple shallow notch on the border head in medio-caudal direction | Small and shallow, oriented more medially |
| | Greater trochanter | Small and developed in the cranial direction | Large and developed in the latero-distal direction |
| | Lesser trochanter | Thin and bordering the caudal border of the shaft medial side | Thick, occupying the whole width of the medial side |
| | Lines on the cranial side | Medial line running straight along the side | Medial line strongly concave along the side |
| | Third trochanter | Rounded and poorly developed | Strong and developed toward the greater trochanter |
| | Trochlea | Oriented medially with a shallow groove and developed medial lip | Oriented cranially with a deep groove and an extremely developed medial lip |
| | Condyles | Almost of the same size | Medial condyle more developed than the lateral one |
| | Intercondylar space | Wide | Narrow |
| T | General aspect | Gracile | Robust |
| | Proximal condyles | Nearly equal surface areas; lateral condyle more developed caudally with a sliding surface for the *m. popliteus* | Medial condyle surface twice as wide as the lateral one and more developed caudally |
| | Intercondylar tubercles | Nearly of equal height | Medial tubercle higher than the lateral one |
| | Central intercondylar area | Wide | Narrow |
| | Tibial tuberosity | Laterally deviated | Massive and oriented in lateral direction |
| | Tuberosity groove | Deep | Shallow |
| | Extensor sulcus | Shallow | Shallow |
| | Proximal articular surface for the fibula | Nail-shaped | Triangular |
| | Interosseous crest | Sharp | Smooth |
| | Distal articular surface for the fibula | Narrow and triangular | Wide and triangular |
| | Articular surface for the talus | Rectangular, slightly tilted laterally | Squared, slightly oriented medially |
| | Medial groove for the talus | Deep and narrow | Deep and narrow |
| | Lateral groove for the talus | Shallow and wide | Shallow and wide |
| Fi | General aspect | Gracile | Robust |
| | Head | Flat and large, oriented cranio-medially | Small and oriented cranially |
| | Proximal articular surface for the tibia | Nail-shaped | Triangular |
| | Shaft | Thin and slightly concave, with two sharp crests running along the lateral side | Broad and straight, with two smooth crests running along the lateral side |
| | Distal articular surface for the tibia | Triangular, narrow and long | Triangular, wide and short |
| | Lateral malleolus | Two well-developed tubercles caudally oriented and separated by a deep groove | Two flat tubercles laterally oriented, with the cranial one being more developed, and separated by a shallow groove |
| | Articular surface for the talus | Kidney-shaped, broad in proximo-distal direction | Triangular, proximo-distally compressed |

**Note:**
B, bone; Fe, femur; Fi, fibula; T, tibia.

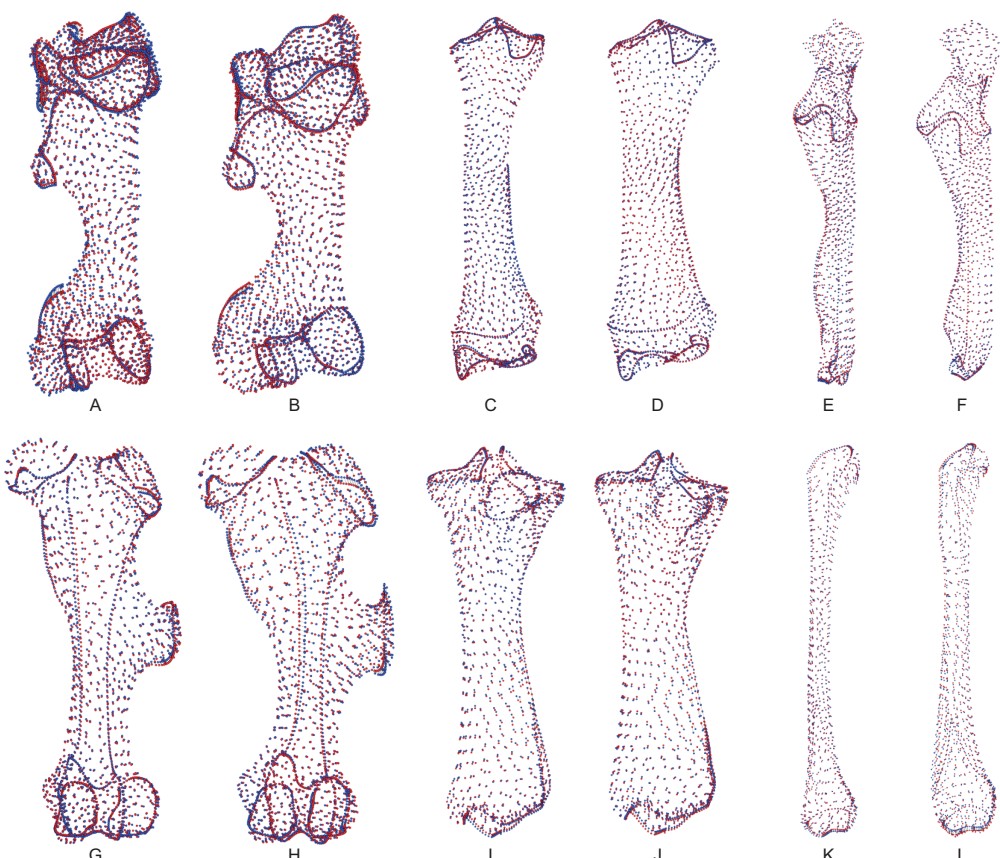

**Figure 8 Landmark conformations associated with minimal and maximal centroid size and mean mass for each bone.** (A, B) Humerus (caudal view); (C, D) radius (dorsal view); (E, F) ulna (dorsal view); (G, H) femur (cranial view); (I, J) tibia (cranial view); (K, L) fibula (lateral view). Red dots, landmark conformation associated with the mean mass. Blue dots, landmark conformation associated with the centroid size. (A, C, E, G, I, K) Landmark conformation associated with the minimum of both parameters; (B, D, F, H, J, L) landmark conformation associated with the maximum of both parameters.

log centroid size. The humerus, the radius and the femur display the highest coefficients, between 0.26 and 0.33. These higher values may be due to the use of a same mean body mass for each rhino species instead of individual mass. Moreover, group affiliation could not be used in this case because of the mean body mass redundancy.

Multivariate regressions of shape scores against log-transformed centroid size (Fig. 9) show that *Dicerorhinus sumatrensis* has the smallest centroid size and is well separated from the other rhino species in most cases, except for the tibia and fibula. *R. unicornis* possesses the highest centroid size in most of the cases, except for the radius and ulna, where it shares similar centroid size values and shape scores as *C. simum* (Table 8). Different tendencies can be observed: for the humerus, Asiatic rhinos have lower shape scores than African ones for a given size. Radius and ulna data display a point pattern similar to each other, with the isolation of *Dicerorhinus sumatrensis* toward low values, a second cluster formed by *Diceros bicornis* and *R. sondaicus* at average values, and a third cluster with *C. simum* and *R. unicornis* showing the highest values. This separation in three

**Table 6 Results of the Procrustes ANOVA performed on shape data and log-transformed centroid size (Cs.) taking into account species (Sp.) affiliation.**

|  |  | $R^2$ | $F$ | $Z$ | $P (> F)$ |
|---|---|---|---|---|---|
| Humerus | Cs. | 0.13 | 17.38 | 5.13 | **0.001** |
|  | Sp. | 0.53 | 17.72 | 8.50 | **0.001** |
| Radius | Cs. | 0.18 | 15.72 | 5.74 | **0.001** |
|  | Sp. | 0.32 | 7.07 | 8.83 | **0.001** |
| Ulna | Cs. | 0.16 | 12.94 | 6.19 | **0.001** |
|  | Sp. | 0.36 | 7.31 | 9.27 | **0.001** |
| Femur | Cs. | 0.14 | 14.41 | 6.07 | **0.001** |
|  | Sp. | 0.37 | 9.56 | 10.08 | **0.001** |
| Tibia | Cs. | 0.13 | 11.62 | 5.13 | **0.001** |
|  | Sp. | 0.36 | 8.06 | 9.03 | **0.001** |
| Fibula | Cs. | 0.10 | 6.61 | 3.77 | **0.001** |
|  | Sp. | 0.26 | 4.47 | 5.61 | **0.001** |

Notes:
$R^2$, determination coefficient value; $F$, Fisher distribution value; $Z$, normal distribution value; $P$, $p$-value.
Significant results are indicated in bold.

**Table 7 Results of the Procrustes ANOVA performed on shape data and cube root of the mean body mass.**

|  | $R^2$ | $F$ | $Z$ | $P (> F)$ |
|---|---|---|---|---|
| Humerus | 0.33 | 25.664 | 5.73 | **0.001** |
| Radius | 0.29 | 18.77 | 6.06 | **0.001** |
| Ulna | 0.21 | 11.22 | 5.57 | **0.001** |
| Femur | 0.26 | 18.61 | 6.39 | **0.001** |
| Tibia | 0.18 | 11.16 | 5.50 | **0.001** |
| Fibula | 0.11 | 5.91 | 3.40 | **0.001** |

Notes:
$R^2$, determination coefficient value; $F$, Fisher distribution value; $Z$, normal distribution value; $P$, $p$-value.
Significant results are indicated in bold.

groups can be observed at a lesser extent for the femur, where *Diceros bicornis* and *R. sondaicus* share almost the same centroid size and shape score variations, whereas *C. simum* and *R. unicornis* are separated by their respective centroid size despite similar shape scores. Finally, tibia and fibula display rather similar patterns with an important intraspecific shape variation, notably for *Dicerorhinus sumatrensis* and *Diceros bicornis*. There is a more important continuity between the different clusters for the tibia and the fibula than for other bones, where clusters are more separated from each other.

## DISCUSSION

### Identification of morphotypes and phylogenetic influence

Morphological variation isolates each rhino species from the others, more or less clearly depending on the bone considered. The shape analysis of the six bones allows for clear isolation of three general bone morphotypes: the African morphotype grouping *C. simum*

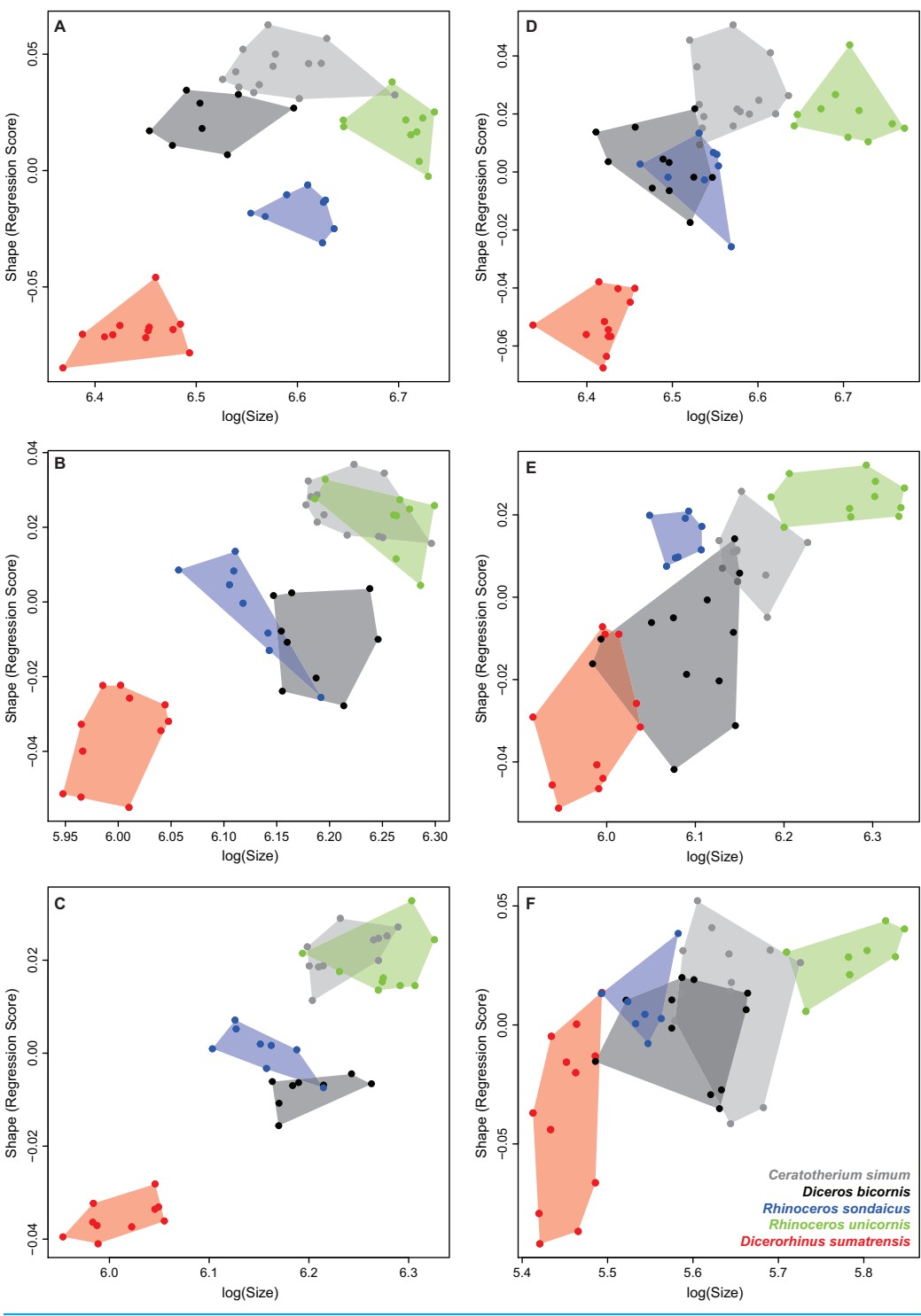

**Figure 9 Multivariate regression plots performed on shape data and log-transformed centroid size.** (A) Humerus; (B) Radius; (C) Ulna; (D) Femur; (E) Tibia; (F) Fibula.

and *Diceros bicornis*, the *Rhinoceros* morphotype grouping the two *Rhinoceros* species, and the *Dicerorhinus sumatrensis* morphotype. The congruence of these morphotypes with the phylogeny indicates that the phylogenetic signal on long bone shape is strong, although

**Table 8 Mean centroid size and standard deviation by bone for each species.**

|          | C. simum | D. sumatrensis | D. bicornis | R. sondaicus | R. unicornis |
|----------|----------|----------------|-------------|--------------|--------------|
| Humerus  | 723 ± 34 | 626 ± 24       | 660 ± 49    | 749 ± 39     | 812 ± 26     |
| Radius   | 501 ± 19 | 403 ± 14       | 485 ± 19    | 463 ± 28     | 520 ± 21     |
| Ulna     | 512 ± 18 | 408 ± 14       | 492 ± 18    | 478 ± 28     | 530 ± 22     |
| Femur    | 724 ± 37 | 613 ± 18       | 657 ± 28    | 686 ± 22     | 822 ± 34     |
| Tibia    | 471 ± 17 | 398 ± 15       | 442 ± 25    | 451 ± 39     | 535 ± 28     |
| Fibula   | 279 ± 14 | 233 ± 7        | 269 ± 14    | 254 ± 8      | 327 ± 16     |

it fluctuates among bones. In addition, body mass also appears as an important factor, depending on the considered bones. The phylogeny is clearly the main effect driving the shape(s) of the humerus and femur. Conversely, the morphological variation observed on the radius and ulna is essentially associated with body mass. The tibia seems to be equally affected by both, which is also the case for the fibula that shows, in addition, an important intraspecific variation.

Despite the fact that we could not test the phylogenetic signal in our data because of the small number of studied species (*Adams, 2014*), our observations tend to indicate an effect of phylogenetic relations. It is accepted that the two African rhino *C. simum* and *Diceros bicornis* are closely related (*Tougard et al., 2001*). They may belong to the same subfamily—called Dicerotinae (*Guérin, 1982*; *Gaudry, 2017*) or Rhinocerotinae (*Antoine, 2002*; *Becker, Antoine & Maridet, 2013*), depending on the authors. The two species composing the genus *Rhinoceros* are also closely related (*Tougard et al., 2001*); the bones of *R. unicornis* and *R. sondaicus* having sometimes been confused with each other (*Groves & Leslie, 2011*). Conversely, the phylogenetic position of *Dicerorhinus sumatrensis* remains debated (*Willerslev et al., 2009*; *Gaudry, 2017*), this species being considered alternately as sister taxon of the two African species (*Antoine, Duranthon & Welcomme, 2003*; *Cappellini et al., 2018*), of the two *Rhinoceros* species (*Tougard et al., 2001*; *Welker et al., 2017*) or of all four other rhino species (*Fernando et al., 2006*; *Piras et al., 2010*). Our analyses reveal equally contrasting relationship patterns, with *Dicerorhinus sumatrensis* more closely resembling African species for some bones (radius, ulna and tibia) and Asiatic ones for the others (humerus, femur and fibula).

Some anatomical features seem strongly influenced by phylogenetic relationships, among which some have previously been used as characters for cladistics analyses (*Prothero, Manning & Hanson, 1986*; *Cerdeño, 1995*; *Antoine, 2002*). On the humerus, the bicipital groove allows the sliding of a large *m. biceps brachii*, a forearm flexor playing an important locomotor role in coordinating the scapula and arm movements (*Watson & Wilson, 2007*; *Barone, 2010b*). This groove appears more closed by the greater tubercle for Asiatic rhinos, potentially indicating a different length and shape for the transverse humeral ligament. Although most analyses (*Prothero, Manning & Hanson, 1986*; *Antoine, 2002*) have coded a few characters related to the tubercles of the humerus, the complexity of the shape of this bone proximal epiphysis remains generally underestimated in phylogenetic reconstructions. Moreover, the case of the greater tubercle development

observed on the humerus of Asiatic species, and mainly for *Dicerorhinus sumatrensis*, is of particular interest (see Fig. S5). As mentioned by *Hermanson & MacFadden (1992)*, the greater tubercle "increases mechanical advantages" for the *mm. pectoralis ascendens*, *supraspinatus* and *infraspinatus*. *Dicerorhinus sumatrensis* displays the slenderest humerus of all modern rhinos, with morphological traits qualitatively close to tapirs' (*MacLaren & Nauwelaerts, 2016*). The proximal epiphysis of *Dicerorhinus sumatrensis* resembles that of tapirs, regarded by some authors as a plesiomorphic condition among Perissodactyla (*Prothero, Manning & Hanson, 1986*; *Hermanson & MacFadden, 1992*; *Antoine, 2002*). This particular shape may thus represent an evolutionary heritage and it is unclear whether and how functional constraints may have also affected this shape. The greater tubercle being also an insertion area for the *m. supraspinatus*, extension movements thus seem achieved differently between African and Asiatic rhinos. *Watson & Wilson (2007)* showed that the *m. supraspinatus* in horses acts more as a shoulder stabilizer than as a true extensor of the shoulder. Given the qualitative similarity of shape of this joint between African rhinos and equids, it is likely that this muscle plays a similar role among these groups. The robustness of the lesser trochanter is consistent with a development of the medial end of the *m. supraspinatus*, to increase the shoulder stabilization. The lever arm is medially deflected for *C. simum* and *Diceros bicornis*, and distributed both medially and laterally for *Rhinoceros* species and *Dicerorhinus sumatrensis*. The role of the shoulder joint remains crucial in weight bearing and locomotion, and its shape may be influenced by several factors. The development of a massive greater tubercle is encountered among hippos (*Fisher, Scott & Naples, 2007*), a trait that could be interpreted as an indicator of semi-aquatic habits and displacements into muddy swamps or riverbanks. However, this particular morphology is yet also encountered among domestic bovids, for example (*Barone, 2010a*), which are not semi-aquatic. Conversely, extinct Amynodontidae, presumed to have been semi-aquatic Oligocene rhinos (*Averianov et al., 2017*), did not display this greater tubercle development (*Scott, Jepsen & Wood, 1941*). The development of the greater tubercle can rather be interpreted as an indicator of a powerful shoulder extension, as well as a feature increasing the resistance to displacement on unstable substrates. However, only a comprehensive study of this convergent trait among diverse artiodactyls and perissodactyls taxa could help to understand the functional role of this anatomical region, and its potential link with the ecological habits. On the distal epiphysis, characters related to the shape of the olecranon fossa have been used in phylogenies (*Heissig, 1972*; *Antoine, 2002*). Our results confirm that the shape and depth of this fossa do not seem directly linked to the general bone robustness as observed in these studies. Moreover, this fossa is proximodistally larger for the genus *Rhinoceros* than for *Ceratotherium* and *Diceros*.

On the femur, the *fovea capitis* is extremely reduced in *C. simum* and absent in *Diceros bicornis*, whereas it is well developed in Asiatic rhinos, especially in *R. sondaicus*, confirming previous observations (*Guérin, 1980*; *Antoine, 2002*). This *fovea* provides an attachment for the accessory ligament and the femoral head ligament (*Hermanson & Macfadden, 1996*), acting as a hip stabilizer. The absence or reduction of *fovea capitis* in African species may be both associated with their phylogenetic proximity. This *fovea* is
indeed present in many fossil rhinos (*Antoine, 2002*), regardless of the ecological preferences of these species. The shapes of the greater and of the third trochanters also seem driven more by the phylogeny than by functional constraints, supporting their use in phylogenies (*Cerdeño, 1995*; *Antoine, 2002*). On the distal epiphysis, the medial trochlear ridge is more developed and inflated in all rhinos than in horses; this feature has been previously interpreted as associated with "locking" the knee joint during long standing periods in equids (*Hermanson & Macfadden, 1996*) and considered as functionally equivalent in rhinos (*Shockey, 2001*). Other authors saw in the development of this medial trochlear ridge an adaptation to a more important degree of cursoriality, linked to openness of habitat (*Janis et al., 2012*). But tapirs, yet able to gallop (*Sanborn & Watkins, 1950*), do not display such an enlargement of the medial ridge of the trochlea (*Holbrook, 2001*; C. Mallet, 2019, personal observation). This trait may thus be phylogenetically inherited between horses and rhinos only, or results of a convergence toward a knee-locking apparatus (which has yet to be fully demonstrated for rhinos).

On the tibia, the massive development of the tibial tuberosity seems more pronounced among African species than in Asiatic ones. The angle between the tibial plateau and the shaft axis is interpreted as a functional character linked to the limb posture (*Lessertisseur & Saban, 1967*); a plateau caudally lowered may reflect an angled limb associated with a cursorial habit, whereas a horizontal plateau tends to indicate more columnar limbs. Here, despite a slight change in the plateau orientation between light and heavy rhino species, this trait seems more likely related to phylogeny; African species have a more horizontal plateau than Asiatic ones. Similarly, on the distal epiphysis, the rectangular shape of the articular surface for the talus is encountered mainly in the three Asiatic species and not in African specimens.

## Role of ecology

Phylogenetically related rhinos share ecologies with important similarities, making it difficult to accurately assess the environmental effect on bone shape. Furthermore, as historical ranges and habitats of rhinos have been drastically reduced and modified under human pressure (*Hillman-Smith & Groves, 1994*; *Dinerstein, 2011*; *Groves & Leslie, 2011*; *Rookmaaker & Antoine, 2013*), ecological inferences must be assessed with caution regarding the current rhino habitats. The related *C. simum* and *Diceros bicornis* both live in African savannas and display a common general bone morphotype (see above). *Diceros bicornis* is a ubiquitous species, often visiting both open savannas and clear forests and browsing various vegetal species, whereas *C. simum* is an open grassland grazer (*Dinerstein, 2011*). The same assessment can be done for the two *Rhinoceros* species, closely phylogenetically related and sharing an important part of their historical geographic range. Despite their strong affinity with water, their ecological preferences are quite different, *R. unicornis* feeding frequently in semi-open floodplains whereas *R. sondaicus* prefers denser forests. *R. sondaicus* and *Dicerorhinus sumatrensis* share a similar lifestyle in dense and closed forest habitats but only their humerus, femur and fibula tend to display slight shape similarities. If long bone shape is affected by environmental factors, these constraints are difficult to distinguish from the ones linked to

phylogeny. This tends to confirm previous observations indicating that rhino long bones can hardly be used as accurate environmental markers (*Guérin, 1980*; *Eisenmann & Guérin, 1984*).

## Shape variation, evolutionary allometry and functional implications

Increase in body size and mass between the lightest and heaviest rhinos is associated with a global broadening of the limb long bones, with a clear enlargement of both the diaphysis and epiphyses, confirming previous general observations on different mammalian clades (*Bertram & Biewener, 1990*, *1992*). However, this broadening is not uniform for all the bones. It is directed both mediolaterally and craniocaudally for the humerus (especially for the proximal part), and mainly mediolaterally for the radius and the femur. Conversely, for the ulna, tibia and fibula, we rather observe a craniocaudal enlargement, particularly visible on the proximal part of the tibia.

### Forelimb bones

The difference between high and low size among extant rhinos is expressed on the humerus by a general enlargement in both craniocaudal and mediolateral directions, particularly for the proximal first half. This may be related to the constraints exerted both by weight bearing and braking role of the forelimb during locomotion (*Dutto et al., 2006*). The important development of the lesser tubercle at the expense of the greater tubercle in non-*Dicerorhinus* species allows both a greater stability of the shoulder articulation, preventing hyperextension and a larger insertion area for the medial end of the *m. supraspinatus*, also considered as a shoulder stabilizer (*Fisher, Scott & Naples, 2007*; *Watson & Wilson, 2007*). This muscle being one of the main extensors of the forelimb (*Barone, 2010b*), the developed lesser tubercle acts as a strong medial lever arm for extension movements. This configuration has been previously interpreted as mechanically advantageous for the muscles inserting on the shoulder joint, while the lateral reinforcement of the greater tubercle was supposed to help resisting the adduction of the arm (*Hermanson & MacFadden, 1992*). The development of the lesser tubercle may also help supporting the scapula (more elongated among African rhinos, J. MacLaren, 2019, personal communication) and be associated with a lengthening of the *m. subscapularis* tendons. In addition, the lesser tubercle also displays an important development in *Diceros bicornis*, more pronounced than in *R. unicornis* and *R. sondaicus*, though these species are heavier and taller. This indicates a possible effect of phylogenetic proximity or similar habitats between the African species (see above). The development of the intermediate tubercle for some rhinos may be related to the presence of a forelimb passive stay apparatus, as demonstrated in horses (*Hermanson & MacFadden, 1992*; *Mihlbachler et al., 2014*). Although less developed than in equids, the intermediate tubercle is present in all rhinos at different degrees (well visible in African taxa, less developed in *Rhinoceros* species and poorly developed in *Dicerorhinus*). This may indicate different degrees of development of passive stay mechanism possibly linked to phylogeny and ecology (*Shockey, 2001*). On the distal epiphysis of the humerus, the mediolateral enlargement observed toward high body mass ensures both a greater stability of the elbow

articulation and larger insertion areas for the different flexor and extensor muscles for the digits (*Barone, 2010a*). The distal trochlea of the humerus is also subjected to a proximodistal compression and a mediolateral extension, increasing the articular surface area to dissipate compressive forces, important for maintaining posture at high body masses (*Jenkins, 1973*).

Forelimb paired zeugopodial bones seem to express complementary shape variations linked to body mass. Whereas the radius broadens mainly mediolaterally with increasing body mass, the ulna expands in the craniocaudal direction; they respond conjointly to the increase in body mass and bone size to form a structure reinforced in all directions, as it has been observed on the humerus. All rhinos have an ulnar proximal epiphysis situated caudally to the radius, while its shaft expands laterally, possibly allowing a mediolateral weight distribution. Moreover, almost all the weight is borne by the proximal articular surface of the radius (*Bertram & Biewener, 1992*), which expands medially and becomes asymmetrical for heavier rhinos. The concave radial tuberosity shows a deep *m. biceps brachii* insertion delivering a strong forearm flexion (*Antoine, 2002*) and the developed insertion lateral relief offers a greater surface for the *m. extensor digitorum communis* (*Guérin, 1980*). As this relief is more developed in African species than in Asiatic ones, this may suggest an effect of phylogeny or locomotion in different habitats or both. On the ulna, the developed olecranon process constitutes a powerful lever arm for forearm extensors such as the *m. triceps brachii* and the *m. anconeus*, also acting upon the bone for gravitational support. The medial development of the olecranon process is related to larger insertions for the *mm. flexor carpi ulnaris, flexor digitorum profundus* and *flexor digitorum superficialis*, all essential to resist hyperextension of the wrist. The cranially reduced anconeal process allows a greater extension of the forearm than in other taxa (e.g., bovids or equids) (*Hildebrand, 1974*) but prevents a complete verticality of the member as observed in elephants for example (*Osborn, 1929*). The distal epiphysis shows a reduction of both radial and ulnar styloid processes toward high body mass, adding a mediolateral degree of freedom to the wrist articulation. However, the proximally reduced articular surface for the scaphoid limits the craniocaudal wrist flexion (*Yalden, 1971*). These morphological traits allow the foot to bear the weight on different substrates while limiting the risk of wrist hyperflexion (*Domning, 2002*).

### Hind limb bones

In the hind limb, the femur expands mainly in the mediolateral direction for rhinos with high body mass and bone size, tending to indicate a stronger resistance to constraints both linked to body propulsion and weight bearing (*Lessertisseur & Saban, 1967*), exerted in the mediolateral direction (*Hildebrand, 1974*). The mediolateral reinforcement of the femur is mainly located under the head and the neck, responding to a concomitant enlargement of the medial condyle and epicondyle on the distal epiphysis, both indicating an increase of the body load near the sagittal plane. The more distal location of the lesser trochanter improves the lever arm of the *mm. psoas major* and *iliacus*, developing slower but stronger hip flexions (*Hildebrand, 1974*; *Polly, 2007*). The same phenomenon is observed with the third trochanter, situated half way along shaft—contrary to in cursorial

Perissodactyla like equids, where the third trochanter is more proximally situated (*Hermanson & Macfadden, 1996*; *Holbrook, 2001*; *Barone, 2010a*). However, it has been shown that the relative position of the third trochanter barely varies among extinct rhinoceroses considered as "cursorial" or "semi-cursorial" (*Prothero, 2005*). This position along the shaft may thus be influenced by both mechanical and phylogenetic constraints. The extreme development of the third trochanter associated with a distolateral development of the greater trochanter also creates a large lever arm for the *fascia glutea*, the *mm. gluteus superficialis* and *gluteus medius* allowing strong hip flexion and abduction. This association appears the greatest for *R. unicornis*, where the greater and third trochanters can be fused by a bony bridge. Conversely, the greater trochanter is less proximally developed than in related groups like horses and tapirs (*Radinsky, 1965*; *Hermanson & Macfadden, 1996*; *Holbrook, 2001*); as this trochanter is the insertion area for the *m. gluteus medius*, the main extensor of the hip, the extension in rhinos seems less powerful than in cursorial perissodactyls. On the distal epiphysis, the lateral torsion of the rotation axis of the trochlea in heavy rhinos also indicates a more laterally deviated position of the knee. This conformation may improve weight bearing, shifting the body mass laterally to the body, as previously observed on a study of pressure patterns of the feet in *C. simum* (*Panagiotopoulou, Pataky & Hutchinson, 2019*). No real difference in the bone curvature related to body proportion was noticed, confirming previous observations on the independence of femur curvature with regard to body mass increase in quadrupedal mammals (*Bertram & Biewener, 1992*).

On the hindlimb zeugopodial elements, when the proximal epiphysis of the tibia broadens craniocaudally, the proximal fibular epiphysis is reduced in this direction, despite an increased general robustness. The proximal epiphysis of the fibula is also oriented far more cranially than in lighter specimens. The enlargement of the tibial plateau thus seems to involve a relative reduction in size of the fibular head. The distal epiphyses of both bones covary too, with a broadening mainly expressed in the craniocaudal direction. The medial condyle of the tibial plateau enlarges strongly, resulting into an asymmetrical proximal epiphysis. Moreover, the broadening of the tibial tuberosity correlates with a stronger and larger patellar ligament, reinforcing the knee articulation and therefore the lever arm created by the patella (*Hildebrand, 1974*). On the distal epiphysis, the two malleoli are more mediolaterally inflated but less distally expanded, allowing the tarsal articulation to move more freely in heavier rhinos (*Lessertisseur & Saban, 1967*). This trait is associated with a slightly shallower distal articular surface, conferring more important degrees of freedom to the ankle articulation for high body mass (*Polly, 2007*). This observation is coherent with similar analyses conducted on rhino ankle bones (C. Etienne, 2019, personal communication, showing notably that the talus bone is flattened and has a shallower groove toward high body mass among rhinos).

In addition to the reduction of the proximal epiphysis, the fibula displays a straighter diaphysis for large rhinos as opposed to the greatly curved one for lighter rhinos (see Figs. S6 and S7). This is consistent with previous observations: although the fibula was not considered in their study, *Bertram & Biewener (1992)* noted a decrease of tibia curvature while body mass increases among terrestrial mammals. In our rhino sample, the tibia

shows a very slight straightening of the diaphysis. However, this straightening, perhaps linked to load carrying capacity, appears to be more pronounced on the fibula.

## Differences between body mass and body size

As the exact body mass was only known for five specimens of our sample, we were not able to precisely express the shape variation regarding the animal's individual weight. However, theoretical bone shape obtained with mean body mass are very similar to the ones obtained with centroid size (see above). Comparing the values of the centroid size and mean body mass highlights some interspecific differences: if *Dicerorhinus sumatrensis*, the smallest rhino, has the lowest values for both centroid size and body mass, *R. unicornis* (the species with the highest values of shoulder height) displays the highest values of centroid size in most cases, which is coherent with its higher height at shoulder compared to other modern rhinos (*Guérin, 1980*; *Dinerstein, 2011*), despite a mean body mass (2,000 kg) lower than that of *C. simum* (2,300 kg). Furthermore, the centroid size of an isolated bone may neither reflect the actual global size of an animal, nor be strictly correlated with its body mass. This is particularly visible for taxa displaying brachypodial adaptation (i.e., shortening of limb length relatively to the height at the shoulder), as it is the case for modern hippos or some fossil rhinos like *Brachypotherium* or *Teleoceras* (*Cerdeño, 1998*). However, our results indicate that it does not seem to be the case with the long bones of modern rhinos. As bone size and body mass are intimately entangled (*Berner, 2011*), the centroid size of isolated bones may still constitute a useful body mass approximation when precise body mass remains unknown and if considered cautiously—this approximation depending on the number and placement of the landmarks on the bone. This is coherent with previous results obtained on cranial shape data indicating a marked correlation between body mass and centroid size (both of the skull and mandible) for many mammalian lineages, especially modern rhinos (*Cassini, Vizcaíno & Bargo, 2012*). Another study focusing on tapirs tend to highlight a good correlation between centroid size and body mass estimation when using the forelimb elements (*MacLaren et al., 2018*).

## Limb bone shape and graviportality

One of the criteria defining graviportality is straight and columnar limbs (*Gregory, 1912*; *Osborn, 1929*; *Biewener, 1989b*). Rhino limb long bones do not display a true columnar organization (*Osborn, 1900*, *1929*). Morphological changes between light and heavy rhino species do not imply a clear change in the orientation of the articular facets: the elbow joint remains unable to completely open like the elephant's one and the knee remains markedly angled. Only the humeral proximal epiphysis displays a tenuous orientation change between light and heavy rhinos, allowing a more slightly vertical orientation of this bone for *C. simum* and *R. unicornis*.

Limb straightness can result from the reorientation of the trochlear notch of the ulna in the dorsal direction, allowing an efficient support of the humerus (*Gregory, 1912*), as in proboscideans (*Christiansen, 1999*). Our sample tends to indicate instead that the radius is the main support of the body weight in the forelimb among modern rhinos. The shape of

the radius becomes gradually more robust from light to heavy rhinos, with a strong medial reinforcement of the proximal epiphysis. The particular role of the radius was previously highlighted among a large sample of mammal clades (*Bertram & Biewener, 1992*), its vertical position being parallel to ground reaction forces. This supportive role of the radius is widespread among ungulates and remains of importance even in larger fossil rhinos like Elasmotheriinae (*Antoine, 2002*) and Paraceratheriidae (*Qiu & Wang, 2007*; *Prothero, 2013*). Unlike in elephants, increase in body mass among rhinos is correlated with a more important supportive role of the radius. At the opposite, the ulna's role has not been extensively explored in morphofunctional studies. Our work underlines the complementary role of the ulna relative to the radius, providing more lateral and caudal weight bearing by an enlargement in the dorsopalmar direction. In this regard, the zeugopodial conformation in rhinos is close to the one encountered in hippos (*Fisher, Scott & Naples, 2007*).

Forelimb elements bear more weight than hind limb ones (*Lessertisseur & Saban, 1967*; *Hildebrand, 1974*; *Polly, 2007*) and play an additional braking role during locomotion, particularly proximal elements (*Dutto et al., 2006*). Forelimb bones such as the humerus thus need to be reinforced in all directions in order to support these higher masses in heavier animals. Hind limb bone shape is affected differently than in forelimb by increases in body mass and size. The hind limb bears relatively less weight than the forelimb in quadrupeds and plays an additional propulsive role during locomotion (*Lessertisseur & Saban, 1967*; *Hildebrand, 1974*; *Barone, 2010a*). The femur displays important reinforcement and development of strong lever arms in large rhino species, possibly to support increasing stress due to locomotion and body mass, but the variations in shape of the tibia and the fibula seem driven as much by the body mass as by the phylogenetic influence. The shape of the fibula is particularly variable within several rhino species, questioning its functional role but also the factors driving this strong intraspecific variation. It has been shown that the human fibula plays, in addition to its ankle stabilizer role, a small but important weight bearing role, receiving one sixth of the load applied to the knee (*Lambert, 1971*; *Takebe et al., 1984*). In horses, the diaphysis of the fibula is absent and the malleolus is fused with the tibia, ensuring mainly ankle stabilization (*Barone, 2010a*). The rhino fibula ensures a talus stabilization role (*Polly, 2007*) in addition to a potential weight bearing due to the presence of the shaft. In addition, this bone often bears crests along the diaphysis with no apparent correlation with weight bearing (see above). These crest developments may be due to individual variations in bone development, without clear functional implications, but this first analysis does not allow us to address this question.

*Bertram & Biewener (1990*, *1992*) and *Polly (2007)* previously called "allometry increase" the tendency for body size and mass to rise among terrestrial mammals. Although reduced, this allometry clearly affects our sample (Tables 6 and 7). In addition, robustness increase is associated with a slight relative length reduction of the bone for larger rhinos such as *Ceratotherium* (*Guérin, 1980*), a general trend observed among heavy mammals (*Christiansen, 1999*). Another trait associated with body mass augmentation among extant rhino species is the expansion of the medial epiphyses of multiple bones (e.g., medial

epicondyle and trochlear lip on the humerus, medial glenoid cavity on the radius, medial condyle and trochlear lip on the femur, medial condyle on the tibia). These medial reinforcements result in more asymmetrical bones, potentially increasing parasagittal weight bearing (*Barone, 2010a*). This conformation is coherent with foot posture during walk: rhino forefeet are placed under the body, close to the sagittal plane of the animal (*Paul & Christiansen, 2000*). Hind feet are more spaced and oriented laterally, especially for heavy rhinos (*Pfistermüller, Walzer & Licka, 2011*; *Panagiotopoulou, Pataky & Hutchinson, 2019*), which seems to agree with our observations regarding the rotation axis of the femoral trochlea, oriented more laterally as well. However, the distal articular surface of the tibia displays a broader lateral groove and appears as a counterexample (Fig. 5). This lateral broadening of the ankle joint, also observed on the talus (C. Etienne, 2019, personal communication), may be correlated with the hind limb posture of rhinos. As the pelvic bone is large and the feet are placed under the body and oriented more laterally than forefeet, the legs are not parallel to the sagittal plane (*Paul & Christiansen, 2000*; C. Mallet, 2019, personal observation). The vertical forces exerted by the body mass may therefore cross the axis of the tibia. This appears in accordance with the fact that the forces may be medially higher on the proximal plateau but laterally higher at the ankle joint; this point would need to be tested more precisely in vivo. As studies of pressure patterns indicate that foot pressure is more intense laterally (*Pfistermüller, Walzer & Licka, 2011*; *Panagiotopoulou, Pataky & Hutchinson, 2019*), it will be crucial to explore relations that exist between stylopodium, zeugopodium and autopodium organization in the complete limb, as well as the gait and posture of the rhinos.

## CONCLUSION

This study conducted on the limb long bones among modern rhinos highlights the occurrence of three distinct morphotypes. These reflect phylogenetic relationships, and the bone shape is differently affected by body size and mass. The shape of the stylopodium bones, though affected by body mass variation, remains highly constrained by phylogeny, whereas zeugopodial bones, especially the radius and ulna, are more strongly affected by body mass, which highlights their important role in weight bearing. The shape of the tibia is influenced by both body mass and phylogeny. The unique pattern of the fibula reveals that, beyond significant intraspecific variation, this bone may play a role in weight bearing. Comparisons with hippos and elephants show clear differences and convergences and highlight the interest of investigating shape variation in other heavy mammal taxa. This would enable description of the different ways to sustain an increase of body mass in mammals and, eventually, to sharpen the concept of "graviportality."

## ACKNOWLEDGEMENTS

The authors would like to warmly thank all the curators of the visited institutions for granting access to the studied specimens: Catriona West, Rachel Jennings, Mike Cobb (Powell Cotton Museum, Birchington-on-Sea, UK), Didier Berthet (Centre de Conservation et d'Étude des Collections, Musée des Confluences, Lyon, France), Yves Laurent (Muséum d'Histoire Naturelle de Toulouse, Toulouse, France), Joséphine Lesur,

Aurélie Verguin, Salvador Bailon (Muséum National d'Histoire Naturelle, Paris, France), Roberto Portela-Miguez (Natural History Museum, London, UK), Frank Zachos, Alexander Bibl (Naturhistorisches Museum Wien, Vienna, Austria), Olivier Pauwels, Sébastien Bruaux (Royal Belgian Institute of Natural Sciences, Brussels, Belgium), Emmanuel Gilissen (Royal Museum for Central Africa, Tervuren, Belgium), Anneke H. van Heteren (Zoologische Staatssammlung München, Munich, Germany) and John Hutchinson for providing us CT-scan data coming from the National Museums Scotland (Edinburgh, UK), the University of California Museum of Paleontology (Berkeley, USA) and the University Museum of Zoology Cambridge (Cambridge, UK). C.M. acknowledges Arnaud Delapré (MNHN, Paris, France) for significant help in 3D data reconstruction and management, Cyril Etienne, Rémi Lefebvre, Romain Pintore (MNHN, Paris, France) for constructive discussions and advices on R programming, data analyses and interpretations. We would also like to thank Jamie MacLaren (University of Antwerp, Antwerp, Belgium), Kelsey Stilson (University of Chicago, Chicago, USA) and Luke Holbrook (Rowan University, Glassboro, USA) for their constructive reviews that allowed us to significantly improve the quality of the manuscript.

### Funding

This work was funded by the European Research Council and is part of the GRAVIBONE project (ERC-2016-STG-715300). The funders had no role in study design, data collection and analysis, decision to publish, or preparation of the manuscript.

### Grant Disclosures

The following grant information was disclosed by the authors:
This work was funded by the European Research Council and is part of the GRAVIBONE project (ERC-2016-STG-715300).

### Competing Interests

The authors declare that they have no competing interests.

### Author Contributions

- Christophe Mallet conceived and designed the experiments, performed the experiments, analyzed the data, prepared figures and/or tables, authored or reviewed drafts of the paper, approved the final draft.
- Raphaël Cornette conceived and designed the experiments, performed the experiments, analyzed the data, authored or reviewed drafts of the paper, approved the final draft.
- Guillaume Billet conceived and designed the experiments, authored or reviewed drafts of the paper, approved the final draft.
- Alexandra Houssaye conceived and designed the experiments, authored or reviewed drafts of the paper, approved the final draft.

## Data Availability

The raw data (available in the Supplemental Files) are the spatial coordinates of all the landmarks placed on each specimen. Anatomical and curve landmarks were placed on each specimen, whereas sliding surface semi-landmarks were placed only on the template (suffix "LM_surface" in the file name) and then projected and slided on each specimen.

The R code (available in the Supplemental Files) describes the template creation, sliding process, Procrustes analysis and data analysis for one bone (humerus). Landmark definition is provided for all the other five bones.

Table S2 provides the details regarding the institution, the accession numbers, the available bones, the sex, the age and the condition, as well as the method of 3D acquisition of the data. All the studied specimens are stored in their respective institution.

## Supplemental Information

Supplemental information for this article can be found online at http://dx.doi.org/10.7717/peerj.7647#supplemental-information.

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
