# Peer review of "Interspecific variation in the limb long bones among modern rhinoceroses—extent and drivers"

_PeerJ, doi:10.7717/peerj.7647_

## Round 0.1 · original submission · Major Revisions

We have obtained 3 very constructive and efficient (speedy!) reviews for your MS, and all agree it is very publishable with some amendments, which are diverse but all seem do-able. This constitutes moderate revisions overall. Be sure to address all points indvidually in your rebuttal. I agree this is a great contribution to PeerJ and the scientific literature!!

·

Basic reporting

In the manuscript titled “Interspecific variation in the limb long bones among modern rhinos - extent and drivers”, Mallet and co-authors present a neat study on the long-bones of the upper limb in modern rhinoceroses using a three-dimensional geometric morphometric approach. However, considering the broad literature on limb morphology in relation to biotic and abiotic covariates (e.g. phylogeny, habitat etc.), I am surprised that this study lacks hypotheses on what may be found using a 3D-GM assessment of extant rhinocerotid proximal limb bones. I think this study is missing a clear and testable hypothesis; this, unfortunately, makes the Results section quite laborious to read, as it has no focal point to base the descriptions around. I suggest that even a very simple biogeographic, size-related or phylogeny-based hypothesis will allow the Results and Discussion sections to flow more smoothly towards a focused conclusion.
This article is in general well written and the points are put across in a logical manner. However, I have come across numerous examples of sentences which are in need of restructuring, and words used which are inappropriate (“pending” rather than “depending”, and possessive apostrophes being the chief culprits). There is also an odd combination of American and British English used – this is not a true criticism, but may be something which is in need of consistency throughout the article in accordance with the journal guidelines. I have chosen not to highlight every section in the article which I think is in need of adjustment, but I strongly recommend that the authors have this manuscript read over by a native English speaker before resubmission. The changes themselves will not be major, but there are many instances where a rearrangement of the sentence will allow a much better flow for the reader.
The tables, figures, references and supplementary information are well laid out, logical and easy to understand. There are a few references I feel could be added in key areas - I have made a few suggestions in the attached PDF file.

Experimental design

This article is well suited to the aims and scope of PeerJ; it represents an original research article within the biological sciences, performed to a high technical standard. The authors lay out several issues and knowledge gaps (e.g. the label of "graviportal" and whether rhinoceroses can be considered as such). The authors lay out their reasons for chosing rhinoceroses as a subject group well, with logical arguments. They then accumulated an excellent sample of modern rhinoceros upper limb bones, and have used more computationally intensive methods than were previously available to researchers studying locomotor morphology in these animals. The methodologies used are well laid out and informative, although missing a few details which I have highlighted and commented upon in the PDF. The results are comprehensive in their anatomical detail, if a little exhaustive to read at times. The discussion is good, with sound descriptions and comparisons. I feel the authors can delve deeper into the comparative aspects of different parts of the bone which they mention as key features, and several biomechanical concepts need to be readdressed to fit into the story that is being told. My main criticism of the article is the lack of specific aim and testable hypothesis – this absence bleeds down through the results and discussion sections, making them appear disjointed at times. Ecological, biomechanical and phylogenetic aspects are touched upon, but not really discussed in the scope of Rhinocerotidae evolutionary history. I would like to see more engagement with previous literature sources both recent and classic, and I strongly recommend large swathes of results section be relegated to the supplement to benefit whatever hypothesis the authors choose to go with.

Validity of the findings

The findings of this study most definitely benefit the wider literature. The results for interspecific differences between the different rhinoceros species will be valuable for any future studies based on rhinoceros locomotion, graviportality, or general perissodactyl biology. All data utilised for the study have been provided for the reviewers in well organised supplements or ancillary table and figure files, and from what I can tell the methodologies are robust, statistically sound and controlled. I do have some thoughts about the effect of high dimensionality one the data, and how this might affect the statistical comparisons, but these are minor concerns.
Unfortunately, as there was no specific research question or hypothesis, it is difficult to gague the conclusions. Yes, they are limited to supporting the results presented with no rampant speculation. However, I believe that there are aspects of the results coming from this article which would benefit from informed speculation, without drawing concrete conclusions but positing ideas which could be tested in future studies. I have included suggestions within the attached PDF. With the addition of a testable hypothesis and specific research aim, the results will be easier to interpret, the discussion will be more focused, and the conclusions will benefit.

Additional comments

I applaud the amount of work that has gone into this study, and on the whole I am a big fan of this work. The set-up for examining rhinoceros locomotor morphology in the framework of adaptations for “graviportal” locomotion is, in my opinion, the best place to start - of all the modern “graviportal” mammals, rhinos are perhaps most contentiously labelled so. The sample you have acrued is impressive, and clearly a lot of hard work went into the scanning and landmarking procedures.

Given the title and setup to the article, I had the biggest issue with the results section. The reason is simple - there was no actual reference to the shape of the bones of the species for well over 2/3 of the results. Theoretical shapes at axis minima and maxima are excellent means by which to visually inspect what aspects of morphology are most highly variable in a sample. However, the premise of this article (as I understood it) was to inspect the differences between the five extant taxa. There is not much reporting of that for much of the results, but rather an exhaustive report of what features are shaped differently along each PC axis. That, by itself, is interesting - I am not taking anything away from the work put in to identifying every feature which varies. However, I think that in an article quantifying "interspecific variation in the limb long bones of modern rhinos", I expected comparative results between the species to be the centre-piece of the results. Even as a comparative anatomist, I found reading the first two thirds of the results section quite tough, until the interspecific comparisons section. This issue, I think, can be tied back to lack of testable hypothesis – what question are the authors asking, what did they anticipate to be the answer, and do the data support that hypothesis? Once the hypothesis is decided upon, I think the text will fall into place quite easily, and discussion points/speculation will become easier to justify.

With all that being said, I am of the firm belief that this work is an ideal launchpad from which to test theories on rhinoceros locomotion, and the relationship between size and shape of the appendicular skeleton in so called “graviportal” mammals. Beyond that, it is a valuable window on the quite surprising morphological variation exhibited by the five modern rhinoceros species, a group which have for many years had their postcranial morphology somewhat maligned. I would very much like to see this study published, and I hope that the suggestions I have offered may help it along the way.

I wish the authors the best of luck with this study, however they chose to proceed.

Kind regards,
Dr. Jamie MacLaren

·

Basic reporting

This paper is, for the most part, clear and unambiguous. Each section is well-delineated. The introduction hits the major points of appropriate background for the reader. Methods are not completely described in 'Landmark Digitization' (see attached PDF). Some of the technical definitions and wording is lacking at present as well. Figures are well organized and captioned, and only require a few changes. The raw data is available. Hypothesis is vague. This is more of an exploratory study, but the results, implications, and conclusions are strong.

Experimental design

This paper uses established methods (PCA, sliding semi-landmarks, generalized procrustes analysis, etc.) for a novel system: limb bones of extant rhinos. The methods are sound for the most part (see details in attached document), but lack sufficient detail to replicate. The PCA analysis is clear, but the sliding semi-landmark, the Procrustes superimposition with a spline optimized step, and the use of centroid size needs to be expanded upon. This was the main reason I chose 'minor revision' for the paper.

This paper fills an existing knowledge gap because extant rhino limb bones have never been quantified using such an extensive landmark analysis and with such a large sample size. This paper is an important addition to the limited quantitative morphological analyses of rhinocerotidae.

Validity of the findings

The results are valid and meaningful. The paper opts for a dual statistical and descriptive approach. While there are no clear hypotheses, the conclusions are well stated and link strongly with the introduction. The paper makes clear what is data and what is speculation.

All raw data is provided, but there is no associated R Code. While the script that was used to run the analyses is not required, it would be extremely helpful for data replication. See attached.

·

Basic reporting

The authors present a 3-D morphometric study of the limb bones, specifically the long bones, of extant rhinoceroses. They examine the variation among their samples from the five extant rhinoceros species to assess the effect of body size (using centroid size from their data and reported average body masses for each species) and phylogeny. Their results generally show that there is a strong phylogenetic signal, separating three groups: African rhinos, Rhinoceros, and Dicerorhinus. They also record an effect of body size that interestingly is most pronounced for the radius and ulna. The authors use these data to draw conclusions about the evolution of large body size in rhinoceroses and how rhinocerotid adaptations for large body size compare to those of other large terrestrial mammals. Cursory comparisons with hippopotamuses and elephants suggest different types of adaptations in rhinos as compared to these groups.

Overall, this is an interesting study that provides a quantitative approach to examining variation in bone shape that can be related to phylogeny, body size, and locomotion. With the amount of attention that has been given to the morphology of horses, it is refreshing to see current studies like this one and those of MacLaren and colleagues examining the other members of the Perissodactyla. I am particularly hoping that this study will be followed by examination of interspecific variation in other rhinocerotoids. I recommend publication with minor revision.

Listed below are suggested edits, with numbers referring to lines in the manuscript. Most of these suggestions pertain to issues of language.

Title:
1: Use “rhinoceroses” instead of “rhinos” and follow with “(Mammalia, Perissodactyla)”.

Abstract:
22: “present” instead of “propose”
28: “with a significant effect” instead of “to an important impact”

There are a number of other places in the text where “impact” is used and should be replaced with “effect.” Similarly, there are numerous places where the wrong preposition is used (e.g., “to” instead of “with” or “on” instead of “of”).

29: Delete “the” before “body mass.”
31: “greatest” instead of “maximal”
32: “of” instead of “on”
35: The phrase “reinforcement of the main lever arms” is unclear and probably not appropriate. I am guessing what is being described is having more pronounced attachments for muscles, or perhaps it is just the thickening of long bones to resist stresses produced by muscles. Either way, it is not clear from this phrasing.
37: “depending” instead of “pending;” this error occurs in other places throughout the text.

Introduction
46: “exhibit evolutionary convergence” instead of “present a…trend”
54: Delete “a” before “stylopodium” and “an” before “autopodium.” There are many places in the text where “a/an/the” should be deleted.
63: “extensively” instead of “massively”
76: “the peculiar morphology of hippos” instead of “hippo’s peculiar morphology.”
79: The word “consensual” is used several times in the text; replace it with “agreed upon.” Likewise, “non-consensual” can be replaced with “debated” or something similar.
84: “refined” instead of “sharpened”
86: “some” instead of “ones”
87: “support” instead of “sustain”
91: “maximum” instead of “max”
95: “an aquatic” instead of “a water”
103: “for understanding evolution towards large body mass” instead of “to understand evolution towards a high body mass.”
104: “some” instead of “ones”
105: “explore” instead of “explored”
108: “published” instead of “proposed”
109: “emphasized” instead of “aimed to emphasize”
110: What is meant by “determination criteria?”
111: “variation in” instead of “variations on”
114: “three dimensions” instead of “3D”

Materials and Methods
131: “included” instead of “kept”
143: “sex” instead of “gender”
156: Delete “the” before “Avizo”.
161: “come from” instead of “were borrowed to”
164: “specific” instead of “precise” and “complemented” instead of “completed”
169: “complemented” instead of “completed”
171: “that of hippos” instead of “hippo’s”
194: “procedures such as” instead of “procedure as”
209: What is “preconized”? Perhaps “conceived”?

Results
254: Insert “with” before “very” and “of” before “the diaphysis”.
265: Insert “and” before “a capitulum”.
370: “cluster’s” instead of “clusters”
384: This is one of several places where “expended” is used when “expanded” is meant.
389: “a” instead of “at”
393: “trapezoidal” instead of “trapezoid”
415-416: The sentence starting “D. bicornis” and ending “both axes” is unclear and needs to be fixed.
590: What is meant by “smooth details”?
602: Delete “the” before “group” and “allometry”.
603: Delete “the” before “species”.
604: Delete “the” before “centroid”.

Discussion
631: “allows for clear isolation of” instead of “enables to clearly isolate”
635: Delete “the” at end of line (before “phylogenetic”).
637: Delete “both”.
639: “depending” instead of “pending”
642: “debated” instead of “non-consensual”
674: Insert “the” before “shoulder”.
679: “on the other hand” instead of “at the opposite”
689: Delete period after “trochanters”.
690: “supporting” instead of “endorsing”
693-699: Mihlbachler and colleagues have studied knee locks in rhinos, but I think the only thing in print is an abstract (Shockey et al., Journal of Vertebrate Paleontology vol. 28, supplement to no. 3, p. 142A).
695: Delete “the” before “openness” and “habitat”.
700: “seems” instead of “seem”
703: “with” instead of “to”
704: “a” instead of “an” before “horizontal”
709: Delete “the”.
756 “Paired zeugopodial” instead of “Zeugopodial paired”
757, 760, 762, and 775: “expands” instead of “expends”
760: “caudal” instead of “caudally”
772: “Domning” instead of “Domming”; change this in References as well.
788: “appears greatest” instead of “seems maximal”
789: “On the other hand” instead of “At the opposite”
801: “proximal fibular epiphyss” instead of “one of the fibula one”
803: “fibular” instead of “fibula”
804: “covary” instead of “variate conjointly”
817: Delete “the” before “tibia”.
819: “perhaps” instead of “maybe” and delete “the” before “load”
827: What is meant by “tallest”? Largest centroid value?
841: “is” instead of “are the”
842: “Rhino” instead of “Rhino’s”
860: “ulna’s” instead of “ulna” and “extensively” instead of “extendedly”
870: “the hind” instead of “Hind”
880: “The rhino” instead of “Rhino’s”
881: “to that of the horse” instead of “to the horse’s one”
885: “address” instead of “state on”
887: “for” instead of “to” and insert “to” before “rise”
888: Delete “the” before “robustness”.
890: “with” instead of “to”
898: “to agree” instead of “coherent”
914: Delete “latter”. Replace “also” with “are”.
916: “variation” instead of “variations”
916-918: “whereas zeugopodial bones, especially the radius and ulna, are more strongly affected by body size, which indicates their important role in weight bearing.” instead of “whereas it is more…weight bearing.”
918: “The shape of the tibia is influenced by both body size and phylogeny.” instead of “As for the shape…phylogeny.”
920: “significant” instead of “the important”
921: “Comparisons” instead of “Quick comparisons”
923: “description of” instead of “to describe”

Experimental design

The methods are appropriate, as are the conclusions drawn from the results. My only substantial criticism regards how the authors distinguish “body size,” which appears to be represented by centroid size, from body mass, which is based on average species masses from the literature. I would argue that centroid size and body mass are both proxies for the more abstract concept of “body size.” I would therefore suggest replacing “body size” with “centroid size” whenever the reference is to the empirical observation of centroid size, and replacing references to “body size and body mass” with just “body size,” at least when centroid size and body mass have similar effects on variation.

Validity of the findings

The conclusions are appropriate, though they suggest a need for further comparisons with other large-bodied terrestrial mammals, especially since the authors preface the paper by discussing and emphasizing graviportal adaptations.

---

## Round 0.2 · Minor Revisions

One reviewer suggests some minor rewording throughout. I leave these changes to your discretion but quite a few seem helpful. I look forward to seeing the final MS- please ensure to address all points individually in your response.

·

Basic reporting

In the revised manuscript titled “Interspecific variation in the limb long bones among modern rhinoceroses - extent and drivers”, Mallet and co-authors present a thorough revision of their initial article investigating upper limb in modern rhinoceroses using a three-dimensional geometric morphometric approach. I am happy to see the inclusion of a clear testable hypothesis, which is shown to be partially supported by their Results and built upon in the Discussion and Conclusion. The revised article is well written and the points are put across in a logical manner, with notable improvement to the grammar. Unfortunately, I do not wholly agree with the authors that they have fixed all the problems with regards to wording. I am uncertain whether the authors revisited some of the sentence structure mentioned in my initial review, as there are still several instances where swapping the adjectives and nouns make the sentences make more sense and flow more smoothly – this is especially evident (to me) while reading the Results section. I have offered some suggestions for sentences which I feel are still a bit clunky, or that I feel would benefit from rearrangement (see Comments to Authors); however, I stopped suggestions after the Humerus and Radius sub-sections to prevent an overly laborious list of “suggest change to” comments. I recommend that the authors go over the descriptive sentences in the results and ensure that the adjectives and nouns are in a logical order. I stress that this is not the case for every descriptive feature, and the authors have in general done a very good job of fixing the previous grammatical errors. Many of these adjective-noun suggestions are very minor, and some may simply be personal preference – I defer to the authors’ viewpoints on whether these warrant changing or not.
Beyond the additional grammatical suggestions, I find the tables, figures, references and supplementary information remain well laid out, logical, easy to understand, and any previous comments I made have been addressed. The few key references I felt would assist the article have been incorporated in a concise and appropriate manner, and have improved the background for the study.

Experimental design

At risk of regurgitating my previous review, I feel this article is well suited to the aims and scope of PeerJ as it represents an original research article within the biological sciences, performed to a high technical standard. The authors lay out several issues and knowledge gaps (e.g. the label of "graviportal" and whether rhinoceroses can be considered as such). The authors lay out their reasons for chosing rhinoceroses as a subject group well, with logical arguments. They then accumulated an excellent sample of modern rhinoceros upper limb bones, and have used more computationally intensive methods than were previously available to researchers studying locomotor morphology in these animals. The methodologies used are well laid out and informative, and any missing information from the original article has been provided. The results remain comprehensive in their anatomical detail, and following the rebuttal I now have a beter understanding about why the descriptions are written as they are. The discussion is good, with sound descriptions and comparisons, and I am happy to see the authors delved further into the mechanical implications of the shape variation they observed, including comments on myology alongside osteology. My main criticism of the original article (lack of specific research question and testable hypothesis) has been rectified, and the results and discussion are the better for it.

Validity of the findings

Essentially in repetition of my previous appraisal, the findings of this study most definitely benefit the wider literature. The results for interspecific differences between the different rhinoceros species will be valuable for any future studies based on rhinoceros locomotion, graviportality, or general perissodactyl biology. All data utilised for the study have been provided for the reviewers in well organised supplements or ancillary table and figure files, and from what I can tell the methodologies are robust, statistically sound and controlled. Concerns I voiced about the high dimensionality of the data remain, but I am satisfied with the comments from the authors in the rebuttal such that I no longer think this is an issue for the article itself – more a philosophical issue for dimension reduction in landmark analyses. With the inclusion of a discrete hypothesis, I feel the conclusions are apt and concise. Speculation throughout the Discussion was limited, although comments from the authors in the rebuttal clarified reasoning, for which I am grateful. Overall I believe this is a much improved manuscript, and the findings therein will be of great value in potential future works on graviportality, large mammal locomotion and perissodactyls in general.

Additional comments

I am very pleased to see the authors chose to revise this manuscript – I maintain that it is a very good article with solid arguments for the choice of taxa and methodology, and a well-rounded discussion relating to the results that were obtained. I find this revision a much-improved version of this article. I have some minor comments on content, however the majority of the below comments are suggestions for rearrangement of sentences to enable easier reading, rather than issues with the statements being made.

Introduction – very minor comments and requests
Line 79-80: “However, the peculiar morphology of hippos (barrel-like body and shortened limbs) linked to semi-aquatic habits” – I have seen this mentioned before, but could you include a citation which describes this link?

Line 82-83: “Rhino’s graviportal condition is surely the less agreed upon:” – I feel like this sentence still needs some rearrangement – perhaps something like: The graviportal condition in rhinoceroses is surely the least consensual:

Line 98: “[Indian Rhinoceros]…survive in India…” – Very minor comment, but I believe this species is also extant in Nepal (see IUCN Red List 2008), although I am uncertain whether Dinerstein (2011) also included this population.

Line 135-137: “This effect is supposed to be more pronounced on the stylopodium (humerus and femur) than on the zeugopodium (radius, ulna, tibia and fibula) bones” – I know exactly what you mean, but due to the fickleness of the English language this may be read as the effect “should” be more pronounced (i.e. “is supposed to be”), which is more a statement than a hypothesis. I recommend a very minor alteration for clarity: We predict that this effect will be more pronounced…

Line 139-140: “In addition, we expect a potential effect of the phylogenetic heritage and of the different ecologies of the species.” – The hypothesis is sound, but I think there is too much vagueness in ‘expect a potential effect’, so perhaps just: In addition, we expect an effect of phylogenetic heritage and different species’ ecologies on bone shape.

Material and Methods – no requests, just questions/comments
Line 245-246: “we used a Principal Component Analysis (PCA) to reduce dimensionality” – did the authors use a between-group PCA approach, or a classic one? A very recent preprint study has suggested that bgPCA can hit problems when dealing with high variable numbers (e.g. multiple semi-landmarks), so I thought I would check whether a bgPCA was performed here. If not, please ignore this comment. If it was, perhaps the authors can comment on the rationale behind it, bearing in mind the potential problems. Reference: Cardini, O’Higgins, Rohlf (2019) Seeing distinct groups where there are none: spurious patterns from between-group PCA, bioRxiv pre-print.

Line 253-255: “Neighbour Joining method was used to construct trees based on relative Euclidian distances between individuals based on all principal component scores obtained with the PCA” – I understand the explanation given in the rebuttal, and have no further comment on the article. I do feel however that the greatest benefit of the N-J tree method is to investigate the spread or variation in raw shape coordinates, which as the authors point out is not possible with landmark-heavy datasets. A philosophical point, nothing more!

Results – minor comments, mostly sentence rearrangements
Line 300-301: “a shaft with its maximal width situated between the head neck and the deltoid tuberosity” – I think the authors mean “humeral head” here instead of “head neck”

Line 302: “a distal epiphysis very large because of…” – suggest change to “a very large distal diaphysis due to…”

Line 303-304: “a medial epicondyle mediolaterally wide and craniocaudally compressed” – suggest change to “a mediolaterally wide and craniocaudally compressed medial epicondyle”

Line 304-305: “…compressed olecranon fossa and trochlea, a wide trochlea displaying…” – suggest change to “…compressed olecranon fossa and trochlea; a wide trochlea displaying…”

Line 308: “overhanging caudally the diaphysis.” – suggest change to “overhanging the diaphysis caudally”

Line 310-311: “a slightly marked lesser tubercle convexity whereas the greater tubercle one is massive” – what does “slightly marked” mean in this context? If you mean the convexity is marked (as in markedly obvious), then it stands to reason this cannot be “slight”. Please clarify.

Line 314-315: “a medial epicondyle craniocaudally developed and overhanging the olecranon fossa;” – suggest change to “a craniocaudally developed medial epicondyle overhanging the olecranon fossa”

Line 315-316: “with an axis less dorsoventrally tilted” – suggest change to “with a less dorsoventrally tilted axis”

Line 323: “a distal epiphysis proximodistally extended” – suggest change to “a proximodistally extended distal epiphysis”

Line 326-327: “a distal epiphysis mediolaterally stretched…” – suggest change to “a mediolaterally stretched distal epiphysis…”

Line 342-343: “a protruding lateral insertion relief (i.e. insertion area of the m. extensor digitorum) whereas the radial tuberosity is little prominent” – I have two small comments here; 1) I have found an English text (Budras et al. 2012, Anatomy of the Horse) in which this feature is named the “lateral coronoid process”. 2) there is no muscle (to my knowledge) in the perissodactyl forelimb termed simply the “m. extensor digitorum” – there is the common digital extensor (m. extensor digitorum communis), which originates from the lateral epicondyle and epicondylar ridge of the humerus; and the lateral digital extensor (m. extensor digitorum lateralis), which originates from the lateral epicondyle and (potentially) has some fibres originating at the lateral coronoid process. From my observations however, this process/relief is primarily the site for the insertion of the lateral-collateral ligaments binding the elbow together. I do not think that the term needs changing, but the definition provided by the authors does need revision.

Line 343-344: “a lateral synovial articulation surface for the ulna mediolaterally reduced” – suggest change to “a mediolaterally reduced lateral synovial articulation surface for the ulna”

Line 350-351: “an articular surface for the scaphoid little extended proximally” – suggest change to “a slight proximal extension to the articular surface for the scaphoid”

Line 353-354: “a proximal articular surface less asymmetrical despite the development of the medial part” – suggest change to “a less asymmetrical proximal articular surface, despite the development of the medial part”

Line 355: “a completely flat radial tuberosity” – did you observe any differences in relative size of the radial tuberosity relative to (for example) the radial head? If so, did this relative size difference correlate with size/phylogeny? Just a functional thought…

Line 355-356: “a lateral synovial articulation for the ulna mediolaterally stretched” – suggest change to “a mediolaterally stretched lateral synovial articulation for the ulna”

Line 359-360: “a distal epiphysis far less dorsoventrally compressed and a lateral tubercle on the dorsal side poorly developed” – suggest change to “a less dorsoventrally compressed distal epiphysis and a poorly developed lateral tubercle (on the dorsal side)”

Line 360-362: “a distal articular surface dorsoventrally wide with the surface responding to the scaphoid extending proximally” – suggest change to “a dorsoventrally wide distal articular surface, with the surface corresponding to the proximal extension of the scaphoid”

Line 366-367: “R. unicornis’s extension along the second axis is very limited, contrary to that of C. simum and D. sumatrensis clusters.” – suggest change to “Extension of R. unicornis morphospace occupation along the second axis is very limited, contrary to that of C. simum and D. sumatrensis clusters.”

Line 370-371: “a palmar process opposed to the coronoid process proximally reduced” – suggest change to “a proximally reduced palmar process, opposed to the coronoid process”

Line 371-372: “a distal epiphysis dorsoventrally broad, with a developed lateral prominence” – suggest change to “a dorsoventrally broad distal epiphysis, with a relatively developed lateral prominence”

Line 372-373: “a little developed radial styloid process; an articular surface for the scaphoid proximally extended.”
- Here I will stop making suggested changes for sentences –

Line 383-384: “a massive olecranon tuberosity with a medial tubercle – where inserts the medial head of the m. triceps brachii – oriented dorsally” – an enlarged medial olecranon tubercle also offers greater attachment and mechanical advantage for the ulnar insertion of the superficial and deep digital flexors (involved in maintaining forelimb posture, especially in very large animals). See also Line 393-394.

Line 388: “a broad shaft with a triangular section” – insert “a broad shaft with a triangular cross-section”

Line 390: “(i.e. triquetral or pyramidal bone)” – the palaeontological name for this (certainly in the USA) is the cuneiform

Line 418-419: “D. bicornis and R. unicornis specimens overlap a significant part of the cluster of C. simum” – very nit-picky comment, but may be worth avoiding “significant” in this context in favour of “substantial”, just in case a reader assumes that there was a test for significant difference between morphospace occupations.

Line 424: “concave aspect to the diaphysis axis” – suggest removing “axis”.

Line 553-554: “forming two distal tips responding to the two lateral tubercles” – two distal tips corresponding (?) to…

Line 656-657: “associated to” – there are several instances of this phrase, where I believe the authors mean “associated with”

Line 675: “Linear regressions of shape scores against log-transformed centroid size (Fig. 9)” – here I think you mean multivariate regressions

Discussion – minor content comments
Line 741: “resembles the tapirs’ one,” – suggest “resembles that of tapirs,”

Line 747-750: “Watson & Wilson (2007) showed that the m. supraspinatus in horses acts more as a shoulder stabilizer than as a true extensor of the shoulder. Given the qualitative similarity of shape of this joint between African rhinos and equids, it is likely that this muscle plays a similar role among these groups.” – Would the authors therefore also speculate that the supraspinatus of Asiatic rhinoceroses (with large greater tubercles forming a deep bicipital groove acts in a similar manner to that of tapirs or suids (which also exhibit a very large greater tubercle; Barone 2000), and likely maintains an essential extension component rather than primarily a support role (as in hippopotamids; Fisher et al. 2007)?

Line 750-751: “The robustness of the lesser trochanter is consistent with a development of the medial part of the m. supraspinatus, to increase the shoulder stabilisation.” – what do the authors mean by “a development of the medial part of the m. supraspinatus”? Do they refer to the supraspinatus insertion site? The medial supraspinatus is, technically, attached to the scapula. Please clarify.

Line 755-761: “The development of a massive greater tubercle is encountered among hippos (Fisher, Scott & Naples, 2007) and may be interpreted as a direct link with semi-aquatic habits and displacements into muddy swamps or riverbanks. This particular morphology is yet also encountered among domestic bovids for example (Barone, 2010a), which are not semi-aquatic. Conversely, extinct Amynodontidae, presumed to have been semi-aquatic Oligocene rhinos (Averianov et al., 2017), did not display this greater tubercle development (Scott & Jepsen, 1941).” – The authors mention in their rebuttal that “in the current state of knowledge, it seems difficult to hypothesise a valuable reason for this convergence”, and yet they then make a direct link between this feature and semi-aquatic lifestyle. I feel that their secondary theory of the enlarged greater tubercle as a potential indicator of shoulder extension/support in resistance to displacement on compliant substrates (useful for suids, hippopotamids, tapirs, dicerorhinines etc.) is a more logical step. Perhaps soften the semi-aquatic angle in favour of the compliant substrate theory, and maybe add a remark on what future work could be focussed upon in order to clear up this curious convergence (as provided in the rebuttal).

Line 800-801: “this trait seems more likely related to phylogeny, African species having a more horizontal plateau than African ones.” – there is something amiss here. I suggest including a semi-colon (;) after “phylogeny”, and then one of the “African” should probably change to “Asiatic”

Line 840: “a larger insertion area for the medial head of the m. supraspinatus” – there is no medial head of the supraspinatus as such, as it is a single-belly muscle. Perhaps modify this to “a larger medial insertion area for the m. supraspinatus”

Line 843-844: “This configuration has been previously interpreted as a mechanical advantage for muscles inserting on the shoulder” – this sentence is not quite right; either the configuration “confers increased mechanical advantage for the muscles”, or it is “mechanically advantageous for the muscles” – neither of these are wrong, but the current text does not really make sense.

Line 846-847: “The development of the lesser trochanter may also help supporting the scapula…” – I think the authors mean the lesser tubercle, not trochanter.

Line 872: “possibly allowing a mediolateral weight display” – weight distribution (?)

Line 876: “the developed insertion lateral relief offers a greater surface for extensor muscles of the digits (Guérin, 1980).” – as mentioned above, only the lateral digital extensor and (possibly) the radial carpal extensor pass over this section of the radius; the common digital extensor does not interact with this portion of the radius (as suggested by Guerin 1980). I realise that this is not in agreement with the cited literature; as a compromise, may I suggest the following: “the developed insertion lateral relief offers a greater surface for digit and carpal extensor muscle attachment (Guerin, 1980)”

Line 881-882: “The medial development of the olecranon process is related to larger insertions for the mm. flexor carpi radialis,” – I see here that my previous comment was incorrect. The flexor carpi radialis does not originate on the olecranon, the flexor carpi ULNARIS does. My apologies!

Line 966-969: “As bone size and body mass are intimately entangled (Berner, 2011), the centroid size of isolated bones may still constitute a useful body mass approximation when precise body mass remains unknown and if considered cautiously – this approximation depending on the number and placement of the landmarks on the bone.” – I agree. I will take this opportunity to direct the authors toward my own study from 2018 as an example of bone centroid size correlating with body mass, investigated with respect to GM of appendicular bones.
Reference: MacLaren, Hulbert, Wallace, Nauwelaerts (2018) A morphometric analysis of the forelimb in the genus Tapirus (Perissodactyla: Tapiridae) reveals influences of habitat, phylogeny and size through time and across geographical space. Zool J Linn Soc. 184, 499-515.

Line 982: “Limb straightness can results from the reorientation” – either “can result from” or “results from”

Once again, I congratulate the authors on the work they have done here, and I look forward to seeing this article published in PeerJ. I wish the authors the best of luck with corrections.

All the best,
Dr. Jamie MacLaren

·

Basic reporting

The authors have incorporated all the feedback from the previous round of reviews. The paper is clear and unambiguous, with precise anatomical terms relevant to Rhinocerotidae. The literature and references provide appropriate and sufficient contest, with a clear professional structure. The R Script is clear and readable, and clearly correlates with the raw data. The paper is self-contained, well written, and a clear organization.

Experimental design

The experimental design is clear and expands upon accepted geometric morphometric data collection and analysis. No one has ever made as detailed a morphometric analysis of extant rhino anterior and posterior stylopod and zeugopod bones. Data analysis is clear and the methods are described in sufficient detail to replicate.

Validity of the findings

The findings are valid and carefully considered by the authors. A study on the exact same specimens would be easy to plan with the information given in this work. The paper also clearly adds to modern rhino morphological and functional studies, as well as fossil rhino studies. Habitat as a variable is carefully considered and found to be inconclusive, which is also an important benefit to the literature.

All underlying data have been provided and are statistically sound and controlled.

The conclusions are well stated and link directly to the original research question. Any speculation is clear.

Additional comments

Thank you for allowing me to read and edit this wonderful paper. I am so happy it is resubmitted. It is much stronger now and I was genuinely intrigued while reading the entire piece. I can't wait for it to come out so I can share it with my colleagues.

All the best.

·

Basic reporting

The revisions by the authors satisfy my few concerns regarding the original submission.

Experimental design

The revisions by the authors satisfy my few concerns regarding the original submission.

Validity of the findings

The revisions by the authors satisfy my few concerns regarding the original submission.

Additional comments

The revisions by the authors satisfy my few concerns regarding the original submission.

---

## Round 0.3 · accepted · Accept

Thank you for your very diligent revisions, and for including your R script! I think you've done an excellent job attending to the last reviews. Please seriously consider publishing your peer review history as these are a good example of constructive reviews. Congratulations on acceptance!